# The dorsal and ventral hippocampus contribute differentially to spatial working memory and spatial coding in the prefrontal cortex

Susanne S. Babl¤, Torfi Sigurdsson [ID]*

Institute of Neurophysiology, Goethe University, Frankfurt, Germany

¤ Current address: Ernst Strüngmann Institute for Neuroscience, Frankfurt, Germany
* sigurdsson@em.uni-frankfurt.de

## Abstract

The hippocampus (HPC) supports spatial working memory (SWM) through its interactions with the prefrontal cortex (PFC). However, it is not clear whether and how the dorsal (dHPC) and ventral (vHPC) poles of the HPC make distinct contributions to SWM and whether they differentially influence the PFC. To address this question, we optogenetically silenced the dHPC or the vHPC while simultaneously recording from the PFC of mice performing a SWM task. We found that whereas both HPC subregions were necessary during the encoding phase of the task, only the dHPC was necessary during the choice phase. Unexpectedly, silencing of either subregion did not affect PFC neurons' ability to represent the animal's position, but did alter how it was represented. In contrast, only silencing of the vHPC affected their coding of spatial goals. These results thus reveal distinct contributions of the dorsal and ventral HPC poles to SWM and the coding of behaviorally relevant spatial information by PFC neurons.

## Introduction

The hippocampus (HPC) plays a key role in representing animals' position in space as well as in supporting spatial learning and behavior [1–3]. In rodents, the HPC extends from an anterior dorsal pole to a posterior ventral pole and along this dorsoventral axis (which is anatomically homologous to the posteroanterior axis of the HPC in primates, including humans), HPC neurons display systematic variation in their molecular profile, anatomical connectivity and physiological properties [4–13]. Partly based on these differences, it has been proposed that the function of the hippocampus varies along its dorsoventral axis, specifically that the anterior or dorsal HPC (dHPC) is primarily involved in spatial processing and navigation whereas the posterior or ventral HPC (vHPC) is more involved in emotional behavior [7,8]. However, vHPC neurons also encode the animal's position in space, although they do so less precisely than dHPC neurons [5,9,10]. Furthermore, the vHPC is necessary for

**Data availability statement:** The data underlying the figures as well as scripts to generate the figures from the data can be found at https://doi.gin.g-node.org/10.12751/g-node.ls2xxj/

**Funding:** The work was funded by the following grants from the Deutsche Forschungsgemeinschaft (DFG; https://www.dfg.de) to T.S.: Collaborative Research Center 1193 (Project B03), ANR-DFG Grant SI1942/3-1 and DFG Research Unit 5159 (project TP3; SI 1942/5-1). The Funders had no role in the study design, data collection and analysis, decision to publish, or preparation of the manuscript.

**Competing interests:** The authors have declared that no competing interests exist.

**Abbreviations :** AAV, adeno-associated virus; ANCOVA, analysis of covariance; dHPC, dorsal hippocampus; HPC, hippocampus; ITI, inter-trial-interval; LFPs, local field potentials; PFC, prefrontal cortex; PSTH, peri-stimulus time histogram; vHPC, ventral hippocampus; MRL, mean resultant length; SWM, spatial working memory.

performance on spatial tasks [14–18]. Overall, this suggests that both dHPC and vHPC are involved in spatial processing and behavior, although their exact contributions are likely different.

The HPC does not support spatial behaviors on its own but rather through interactions with numerous cortical and subcortical structures. In particular, the interactions of the HPC with the prefrontal cortex (PFC) have been well documented [19,20]. These interactions are supported by direct as well as indirect projections from both the dHPC and the vHPC to the PFC [21–28]. HPC-PFC interactions are apparent in the coordination of neural activity between the two structures, as exemplified by phase-locking of PFC neuron firing to hippocampal theta oscillations [29–32], coherence of theta oscillations in the two structures [30–33] and modulation of PFC neuron firing by hippocampal sharp-wave ripples [34–36]. The functional role of HPC-PFC interactions has been most extensively examined in the context of spatial working memory (SWM)—the ability to remember recently visited spatial locations. HPC-PFC interactions are dynamically modulated during SWM tasks, both as a function of task phase and behavioral performance [30–32,37–40]. Importantly, these interactions are required for SWM, since disconnecting the two structures by silencing the HPC of one hemisphere and the PFC of the contralateral hemisphere impairs SWM performance [17,18,41]. Impaired HPC-PFC interactions also correlate with SWM impairments in animal disease models [31,37,42].

However, despite extensive evidence supporting the importance of HPC-PFC interactions during SWM as well as other cognitive functions [43], the precise functional role of these interactions remains elusive. Analyses of the timing of neuronal activity in the two structures suggest that HPC-PFC interactions can reflect the influence of the HPC on the PFC [29,31,34] as well as vice versa [44–47]. Yet exactly how the influence of the HPC manifests itself in the activity of PFC neurons is not well understood. In particular, given the differential roles of the dHPC and vHPC [7,8], it is likely that they influence the PFC in distinct ways. To date, only the contribution of the vHPC to spatial processing in the PFC has been examined [39]. However, it is not known whether and how the contributions of the two hippocampal poles differ.

To address this, in the current study we compared how the dHPC and vHPC contribute to SWM and how they influence PFC neuronal activity. To this end, we combined task phase-specific optogenetic silencing of dHPC and vHPC with simultaneous recordings of PFC activity in mice performing a SWM task in a T-maze. We found that dHPC and vHPC make distinct contributions to SWM, with both subregions being necessary for the encoding of spatial information whereas only the dHPC was necessary when remembered information is used to make a behavioral decision. Although silencing of neither the dHPC or the vHPC impaired the ability of PFC neurons to represent the animals' relative position between the start and goal of the T-maze, it did change the manner in which this information was represented. Furthermore, encoding of to-be-remembered goal locations was impaired by vHPC silencing but not by dHPC silencing. Overall, these results reveal distinct contributions of the dorsal and ventral hippocampal poles to SWM and the representation of space by PFC neurons.

## Results

## Task phase selective inactivation of dorsal or ventral hippocampal subdomains reveals complementary functions in SWM

To examine the role of the dorsal and ventral HPC in SWM, we expressed the inhibitory opsin ArchT [48] coupled with GFP in excitatory neurons of either the dHPC (ArchT-dHPC mice) or the vHPC (ArchT-vHPC mice). Control mice expressed only GFP in either of the two structures (GFP mice). For light delivery, optic fibers were implanted bilaterally above the dHPC or vHPC (Figs 1A–1B and S1A–S1B). In a subset of animals, a moveable bundle of stereotrodes for recording neuronal activity was implanted into the PFC, targeting the prelimbic region (Figs 1A–1B and S1C). Animals were then trained to perform a delayed non-match-to-sample task in a T-maze [49]. Each trial began with a sample phase, in which the mouse could collect a reward at the end of one of the goal arms, while entry to the other goal arm was blocked. The animal then returned to the start box of the maze, where it was confined for a delay phase of 15 s. In the following choice phase, both goal arms were open for entry, but reward was only available in the opposite arm to the one visited in the sample phase (Fig 1C). Entry to the same arm as in the sample phase was counted as a SWM error and was not rewarded.

All three groups were trained on the task without any light delivery and learned at a similar rate (S2A Fig; one-way ANOVA for required training days: $p = 0.18$; one-way ANOVA for performance on last training day: $p = 0.81$). Then, hippocampal involvement in SWM was tested by delivering light to the dHPC or vHPC in half of the trials in a pseudorandom sequence (light-on trials) during either the sample, delay or choice phase. The effect of light delivery on performance was quantified by comparing the percentage of correct choices in light-on and light-off trials. Overall, we observed distinct contributions of the dorsal and ventral hippocampal subdomains in SWM. When light was delivered during the sample phase, performance decreased significantly in ArchT-dHPC mice and ArchT-vHPC mice, but not GFP mice (Fig 1D–1E; $p < 0.001$ for ArchT-dHPC mice, $n = 15$; $p < 0.05$ for ArchT-vHPC mice, $n = 8$ mice; $p = 0.94$ for GFP mice, $n = 7$; Wilcoxon signed-rank test). A two-way ANOVA (light × group) revealed a main effect of light ($p < 0.0001$), group ($p < 0.05$) and a significant light × group interaction ($p < 0.01$). In contrast, light delivery during the choice phase reduced SWM performance only when the dHPC, but not the vHPC, was inhibited (Fig 1J–1K; $p < 0.001$ for ArchT-dHPC, $n = 15$; $p = 0.19$ for ArchT-vHPC, $n = 8$). Performance in GFP mice was not affected ($p = 0.47$, $n = 7$). Accordingly, a two-way ANOVA showed a main effect of light ($p < 0.0001$), group ($p < 0.05$) and a significant interaction ($p < 0.01$). Finally, light delivery during the delay phase did not reduce SWM performance in any of the groups (Fig 1G–1H; Wilcoxon signed-rank test: $p > 0.27$ for all three groups; light × group ANOVA: no significant factors or interaction, all $p > 0.80$).

Thus, the effects of light delivery depended both on the experimental group as well as the SWM phase in which light was delivered. To further confirm this, we performed a 3-way ANOVA (light × group × phase). This revealed a main effect of light ($p < 0.01$) and phase ($p < 0.05$) as well as a group × light interaction ($p < 0.05$), a light × phase interaction ($p < 0.01$) and, notably, a group × light × phase interaction ($p < 0.05$). A main effect of group was not observed ($p = 0.21$) nor a group × phase interaction ($p = 0.23$). To specifically compare the effects of dHPC versus vHPC inhibition on SWM, we calculated the difference in performance between light-off and light-on trials and then compared the difference scores of ArchT-dHPC, ArchT-vHPC and GFP mice for each light condition (Fig 1F, 1I and 1L). This revealed a significantly stronger impact of dHPC compared to vHPC inactivation in the choice phase (Fig 1L, $p < 0.05$, Wilcoxon rank-sum test), but not in the sample phase (Fig 1F, $p = 0.076$).

Further analysis of the animals' behavior revealed that although mice were able to follow the task sequence in all silencing conditions, their running speed in the sample or choice phase was slightly reduced when the dHPC or vHPC was inhibited (S2B and S2D Fig). A 3-way ANOVA (group × light × phase) across ArchT-dHPC and ArchT-vHPC mice in sample light and choice light conditions revealed a main effect of light ($p < 0.05$), but no effect of phase and no phase × light or group × light interaction ($p > 0.26$). This indicates that, in contrast to the observed SWM impairment, the effect on running speed was not specific to any of the SWM phases or to the silencing of any individual hippocampal subdomain. To further examine whether the observed changes in running speed might have influenced the animals' choice accuracy, we

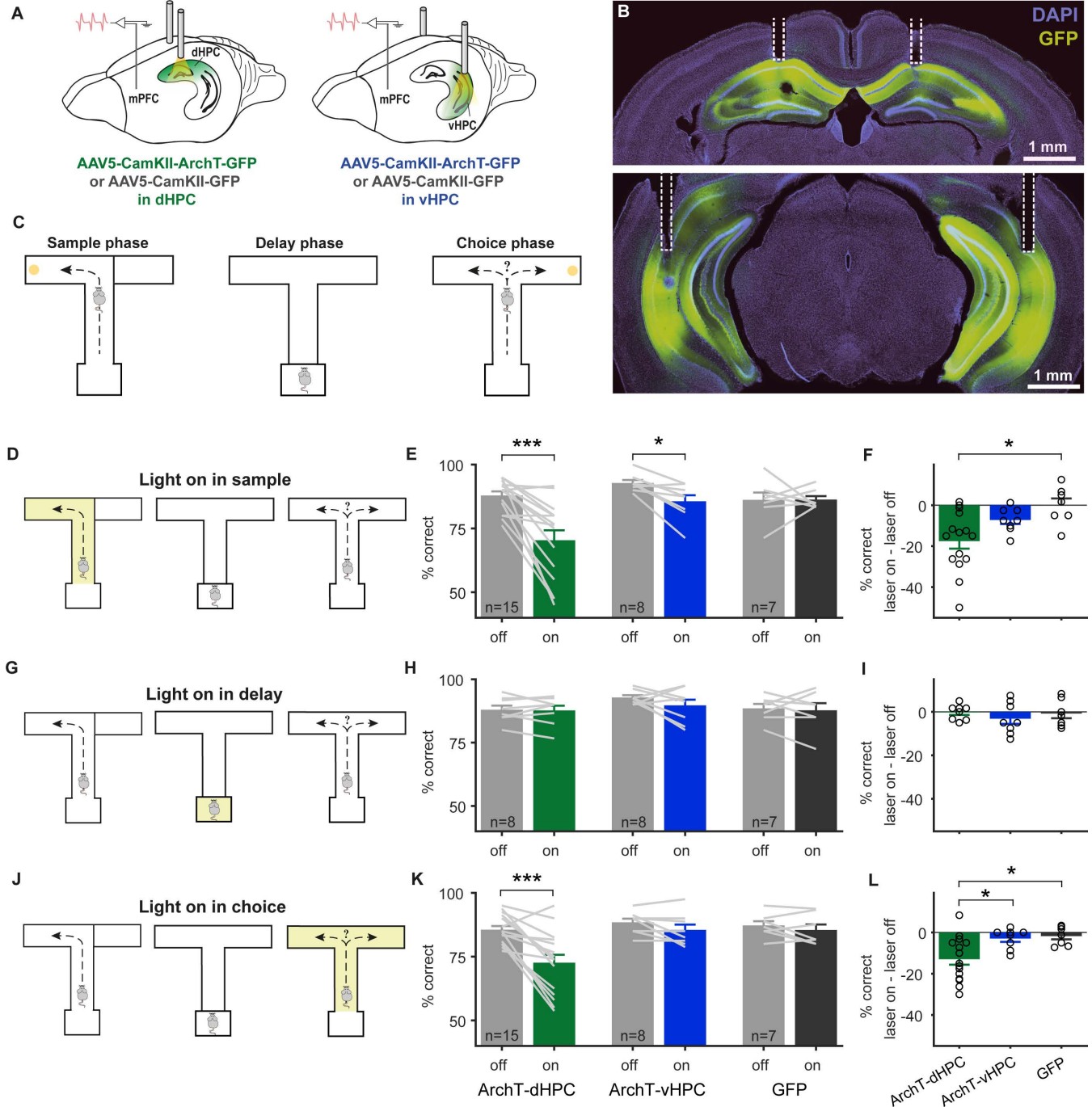

**Fig 1. Optogenetic silencing of dHPC or vHPC during specific phases of spatial working memory. (A)** Schematic of virus injection and optic fiber implantation in dHPC (left) or vHPC (right) with electrodes implanted in the PFC to record neuronal activity during hippocampal silencing. **(B)** Example coronal sections showing ArchT-GFP expression in dHPC (top) and vHPC (bottom) with optic fiber placements outlined by dashed white lines. **(C)** Task schematic. In the sample phase (left), mice were guided to enter one of the two goal arms to receive a reward. After a delay phase in the start box (middle), mice could choose between both goal arms but only received a reward in the goal arm opposite to the one visited in sample (right). **(D–F)** Light delivery during the sample phase **(D)** reduces choice accuracy (% correct) in light-on compared to light-off trials in ArchT-dHPC and ArchT-vHPC mice, but not in GFP control mice **(E)**. Gray lines indicate the choice accuracy of each animal. *$p < 0.05$, ***$p < 0.001$ sign-rank test. Difference scores of light-off versus light-on choice accuracy **(F)** show a significantly larger performance drop in ArchT-dHPC mice compared to GFP mice. *$p < 0.05$, Wilcoxon rank-sum test. **(G–I)** Light delivery during the delay phase **(G)** does not affect choice accuracy **(H, I)**. **(J–L)** Light delivery during the choice phase **(J)** selectively reduces choice accuracy in ArchT-dHPC mice, but not ArchT-vHPC mice or GFP mice **(K)**. ***$p < 0.001$ sign-rank test. Difference scores of

light-off versus light-on choice accuracy **(L)** show a significantly larger performance drop in ArchT-dHPC mice compared to ArchT-vHPC and GFP mice. *$p < 0.05$, Wilcoxon rank-sum test. Error bars represent the mean ± s.e.m. across animals. The number of animals in each experimental group is indicated at the bottom of the light-off bars in **E**, **H** and **K**. The data underlying this figure can be found at https://doi.org/10.12751/g-node.ls2xxj.

performed an analysis of covariance (ANCOVA) using SWM performance in individual sessions as a dependent variable, light as an independent variable and the median speed in each session as covariate. This revealed that the effect of light delivery on SWM performance was not dependent on the animal's speed as there was no significant light × speed interaction in any of the groups ($p > 0.12$; S2E Fig). Additionally, animals did not move slower in the choice phase after dHPC or vHPC inhibition in the sample phase ($p > 0.66$, Wilcoxon signed-rank test; S2B Fig), despite performing more SWM errors (Fig 1D), which further indicates that the SWM deficit and the reduction in running speed reflect two independent effects of HPC silencing on behavior. We also examined whether the animals' performance changed over the course of the testing sessions and whether this could have influenced the effects of HPC silencing. To this end, we performed a light × testing day ANCOVA of SWM performance in individual sessions, separately for each group and silencing condition (S3 Fig). A main effect of testing day was only observed under some conditions (sample light delivery in the ArchT-vHPC group and choice light delivery in the GFP group, $p < 0.05$ and $0.01$, respectively), but in opposite directions. Importantly, a light × testing day interaction was not seen for any of the conditions in which HPC silencing impaired behavioral performance (all $p > 0.57$), or under any other conditions (all $p > 0.20$). These results indicate that the effects of HPC silencing on SWM performance were stable over the period of behavioral testing.

Taken together, these results indicate highly specialized roles of the dorsal and ventral subdomains of the HPC in SWM, with both subdomains being required for the encoding of to-be-remembered information (during the sample phase) but not its maintenance (during the delay phase), whereas only the dorsal HPC is required for using the remembered information to make a behavioral decision (in the choice phase). In contrast, more general behavioral variables such as running speed are influenced similarly by both HPC subdomains.

### Optogenetic silencing of dHPC and vHPC reveals their influence on PFC activity

The HPC supports SWM in part through interactions with other brain regions, notably the prefrontal cortex (PFC), which manifests itself in the coordination of neural activity between the HPC and PFC during SWM tasks [30,31,39]. Analysis of the timing of activity in the two structures suggest that these interactions can reflect the influence of the HPC on the PFC [29,31,34]. We therefore next examined how this influence manifests itself by recording the spontaneous activity of PFC neurons (S4A–S4C Fig) while optogenetically silencing either the dHPC or the vHPC (Fig 2A). To this end, we delivered short light pulses (500 or 1,000 ms, 3 s inter-stimulus interval) while animals explored a small box in the absence of any task demands (Fig 2B). We observed that some PFC neurons in dHPC-ArchT and vHPC-ArchT mice changed their activity in response to light delivery by either increasing or decreasing their firing rate significantly (Fig 2C; $p < 0.05$, sign-rank test, see section "Methods"). Overall, 22.4% of PFC neurons (41 of 183) were modulated by dHPC silencing whereas 28.8% were modulated by vHPC silencing (128/445; Fig 2D). In contrast only 9.6% of neurons were influenced by light delivery in GFP-expressing mice (Fig 2D), a ratio lower than that observed either for dHPC ($p = 0.003$, Fisher's exact test) or vHPC ($p = 3.5 \times 10^{-6}$) inhibition. The overall ratio of modulated neurons did not differ significantly between dHPC and vHPC silencing ($p = 0.11$, Fisher's exact test), but more neurons responded with excitation following vHPC silencing (17.5% or 78 of 445) than dHPC silencing (Fig 2D; 8.7% or 16 of 183; $p = 0.0045$), although the magnitude of firing rate changes were similar when either structure was silenced (Fig 2E). However, vHPC silencing elicited firing rate changes at a shorter latency than dHPC silencing (Fig 2F–G), possibly reflecting the stronger monosynaptic projections from vHPC to PFC [22,23,50]. We also examined responses separately for putative pyramidal (pPYR) and interneurons (pINTs; see section "Methods" and S4D–S4F Fig). This revealed that more pPYRs were modulated by vHPC than dHPC

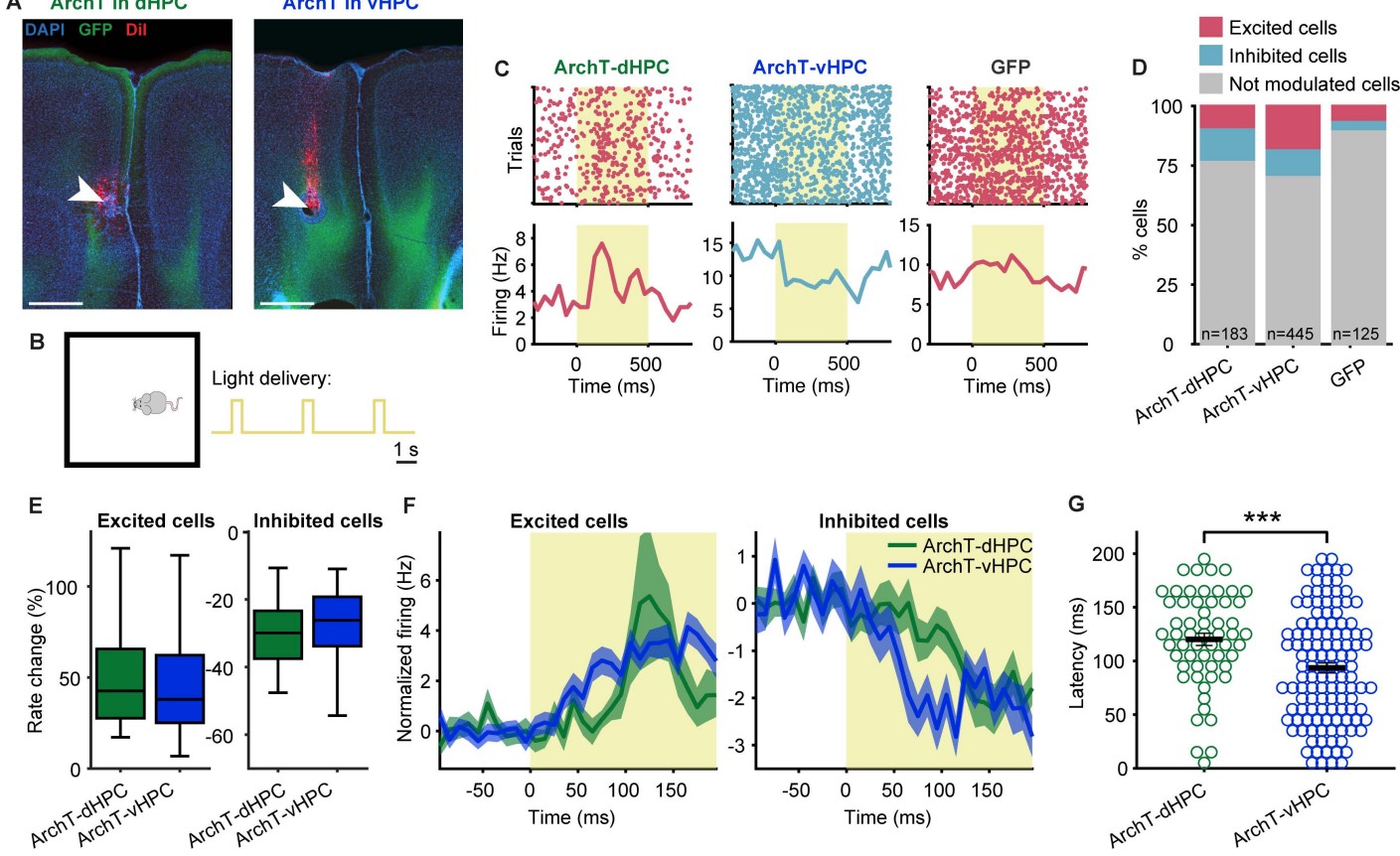

**Fig 2. Brief optogenetic silencing of dHPC or vHPC modulates neuronal firing rates in the PFC. (A)** Example coronal sections showing position of implanted electrodes in PFC (lesion indicated with white arrow, electrode track additionally stained with red DiI) and GFP-positive hippocampal terminals after virus injection and optic fiber implantation in dHPC (left) or vHPC (right). Scale bar: 0.5 mm. **(B)** Schematic of recording session. Mice could move freely in a small enclosure while brief light pulses of 500 ms or 1 s were delivered to the dHPC or vHPC. **(C)** Raster plots (top) and averaged firing rates (bottom) of example cells in PFC showing excitation (red) or inhibition (blue) during light delivery (yellow shaded rectangles) in ArchT-dHPC mice (left), ArchT-vHPC mice (middle) or GFP mice (right). **(D)** Percentages of excited, inhibited and non-modulated cells across groups. More cells were modulated when the dHPC or vHPC was silenced than when light was delivered in GFP mice. The number of recorded neurons in each group is indicated at the bottom of the bars. **(E)** The percent change in firing rate during light delivery relative to baseline in excited (left; ArchT-dHPC *n* = 16 neurons, ArchT-vHPC *n* = 78 neurons) and inhibited (right; ArchT-dHPC *n* = 25 neurons, ArchT-vHPC *n* = 50 neurons) cells was similar in ArchT-dHPC and ArchT-vHPC mice. Box plots represent the median (line), 25th and 75th percentiles (box) and the whiskers extend to the minimum and maximum values within 1.5 times the inter-quartile range below and above the 25th and 75th percentiles, respectively. **(F)** Normalized firing rates (baseline subtracted) around light onset of excited (left) and inhibited (right) PFC cells in ArchT-dHPC mice or ArchT-vHPC mice. Shaded areas indicate mean ± s.e.m. across neurons (*n* values as in **E**). **(G)** Response latencies of modulated cells are longer in ArchT-dHPC mice (*n* = 58) than in ArchT-vHPC mice (*n* = 121). Error bars indicate mean ± s.e.m. across neurons. ***$p < 0.001$, Wilcoxon rank-sum test. The data underlying this figure can be found at https://doi.org/10.12751/g-node.ls2xxj.

silencing ($p$ = 0.002, Fisher's exact test) and that vHPC silencing also modulated more pPYRs than pINTs ($p$ = 0.007; S5A Fig). When restricting our analysis to pPYRs alone (S5B–S5D Fig), we obtained similar results as when using all neurons (Fig 2E–2G). Taken together, these results demonstrate that silencing of either dHPC or vHPC can modulate the spontaneous firing rate of PFC neurons.

## Hippocampal silencing alters the spatial firing patterns of PFC neurons

To better understand how the HPC influences PFC activity, we next examined how silencing of the dHPC and the vHPC affects the encoding of task variables by PFC neurons. Since hippocampal neurons carry information about the animal's

spatial position [1], we reasoned that silencing these neurons would affect the encoding of spatial information in the PFC. To test this possibility, we linearized the animals' position in the T-maze (Fig 3A) and calculated the linearized firing rate of each PFC neuron as the animals ran from the start box to the end of the goal arms of the T-maze during the sample and choice phases (see section "Methods"). The spatial firing patterns of PFC neurons can be highly similar between different goal-directed trajectories, thus forming a generalized representation of the animal's position relative to a goal [51]. For our initial analyses, shown in Figs 3 and 4, we therefore calculated linearized firing rates by combining leftward and rightward trials (see section "Methods"; we analyze these trials separately in Fig 5). Consistent with previous findings [39,49], the activity of many PFC neurons was modulated by the animal's relative position in the T-maze (Fig 3B). To quantify this, we calculated the percentage of neurons showing significant spatial modulation (One-way ANOVA of linearized firing rates, $p < 0.05$, see section "Methods"). Contrary to our expectations we found that this percentage was similar during light-off and light-on trials in both the ArchT-dHPC and ArchT-vHPC groups, regardless of whether light was delivered during the sample phase (Fig 3C; 44% or 187/427 cells during light-off versus 46% or 195/427 cells during light-on in ArchT-dHPC mice, $p = 0.63$, Fisher's exact test; 43% or 170/393 cells during light-off versus 46% or 180/393 during light-on in ArchT-vHPC mice, $p = 0.52$) or the choice phase (Fig 3C; 49% or 215/438 cells during light-off versus 47% or 206/438 cells during light-on in ArchT-dHPC mice, $p = 0.59$; 46% or 155/340 cells during light-off versus 48% or 163/340 cells during light-on in ArchT-vHPC mice, $p = 0.59$). Similarly, silencing of either subregion did not affect the amount of spatial information carried by PFC neurons (Fig 3D). A two-way ANOVA (group × light) revealed no significant main effect of light and no interactions for either sample ($p > 0.26$) or choice light ($p > 0.21$). Similar results were obtained when analyzing pPYRs and pINTs separately (S6A–S6B Fig).

To examine coding of relative position at the population level, we used a Bayesian decoder [52,53] to predict the animal's position from the activity of simultaneously recorded PFC neurons. This enabled above-chance decoding of the animal's linear position on a trial-by-trial basis (Fig 3E) and decoding accuracy increased as expected with the number of simultaneously recorded neurons (Fig 3F). We then compared the performance of the Bayesian decoder when it was trained and tested on light-off or light-on trials. This revealed that the decoding error was not affected by dHPC or vHPC silencing in either the sample or the choice phase (Fig 3G) and neither was the proportion of correctly decoded positions (Fig 3H). A two-way ANOVA (group × light) of either the decoding error or the percentage of correctly decoded positions showed no significant main effect of light and no interactions for light delivery in either the sample ($p > 0.20$) or the choice phase ($p > 0.07$). Taken together, these results demonstrate that silencing the dHPC or the vHPC does not impair the ability of PFC neurons to encode the animal's relative position between start and goal of the T-maze, either at the single-cell or population level.

Although silencing the hippocampus did not alter the ability of PFC neurons to encode the animals' relative position, it might nonetheless have changed *how* they represent this information. Indeed, we observed that some PFC neurons changed their spatial pattern of activity when either the dHPC or vHPC was silenced (Fig 4A–4C). To quantify the similarity of spatial firing patterns with and without hippocampal silencing, we correlated the linearized firing rates between light-off and light-on trials during the task phase in which hippocampal activity was silenced (see section "Methods"). For comparison, we calculated the correlation between the same trials but during the task phase where hippocampal activity was not silenced (e.g., the choice phase in sessions where silencing was performed in the sample phase; see section "Methods"). This revealed that silencing of either the dHPC or the vHPC during the sample phase changed the linearized firing rates of PFC neurons; that is, the correlation between linearized firing rates during light-on and light-off trials was lower in the sample phase than in the choice phase (Fig 4D). A two-way ANOVA (group × phase) revealed a significant main effect of phase ($p < 0.05$), but no effect of group and no interaction ($p > 0.29$). Correlations were lower when either the dHPC or the vHPC was silenced ($p < 0.01$, Wilcoxon signed-rank test) but not when light was delivered in GFP-expressing mice ($p = 0.55$). In contrast, silencing during the choice phase revealed differential effects of the dHPC and vHPC. Silencing of the dHPC, but not the vHPC, altered linearized firing rates, as revealed by lower correlations between light-on and

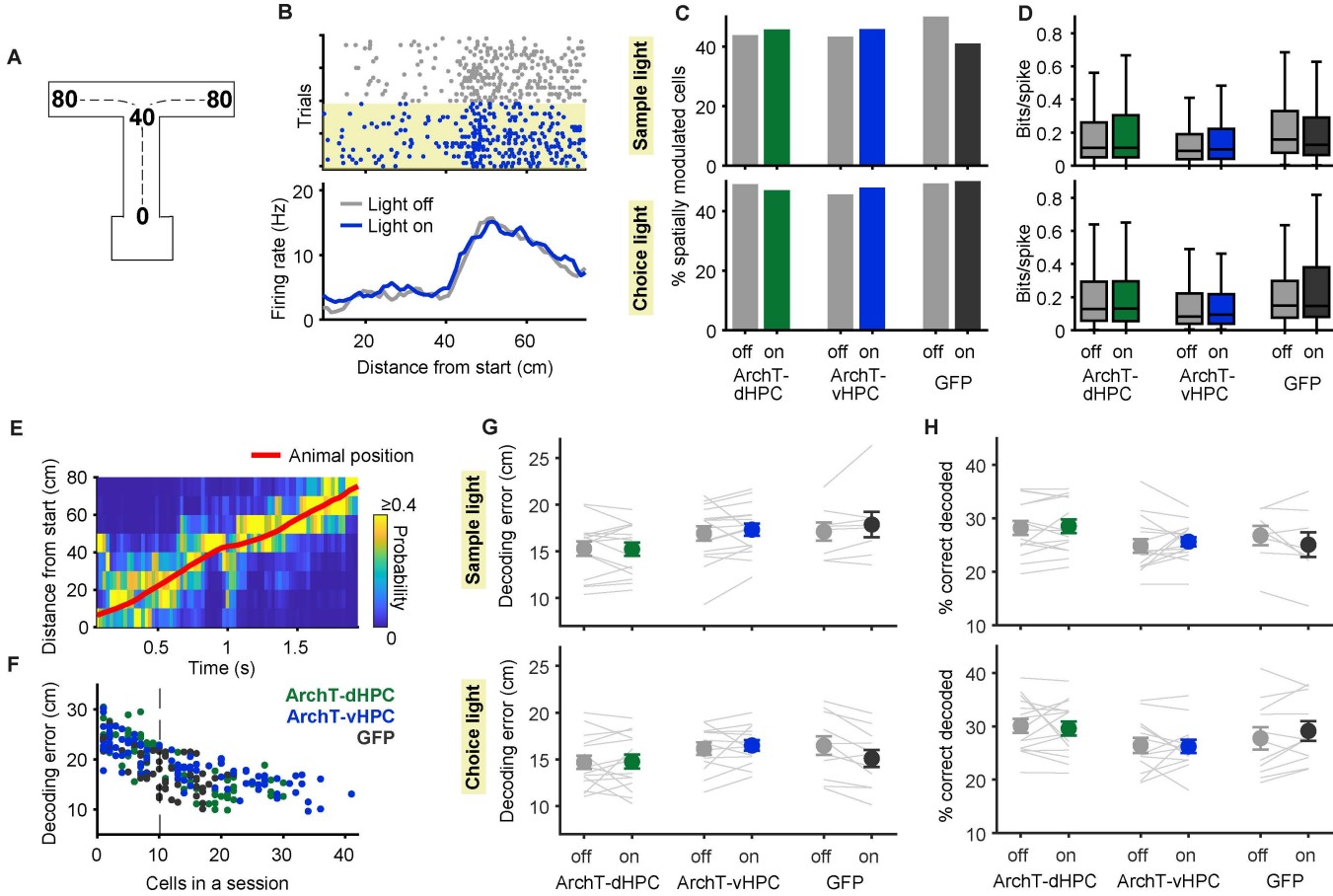

**Fig 3. PFC neurons can encode the animal's spatial position despite dHPC or vHPC silencing. (A)** Schematic showing linearization of the animal's position in the T-maze indicating the distance from the start box in cm. **(B)** Raster plots (top) and averaged firing rate (bottom) of an example neuron as a function of linearized position in the T-maze, as depicted in **A**, during choice phase outbound runs. Note the stable position coding over trials, even during vHPC silencing in the choice phase (yellow shaded rectangle). **(C)** The percentage of PFC neurons significantly modulated by position in the sample phase (top; $n$ = 427, 393 and 229 neurons from ArchT-dHPC, ArchT-vHPC and GFP mice, respectively) and the choice phase (bottom; $n$ = 438, 340 and 227 neurons from ArchT-dHPC, ArchT-vHPC and GFP mice, respectively) is similar in light-off and light-on trials. **(D)** Spatial information (bits/spike) of PFC neurons does not differ between light-off and light-on trials in either the sample phase (top) or the choice phase (bottom). $N$ values as in **C**. Box plots represent the median (line), 25th and 75th percentiles (box) and the whiskers extend to the minimum and maximum values within 1.5 times the interquartile range below and above the 25th and 75th percentiles, respectively. **(E)** Example of Bayesian decoding of linearized position from neuronal population activity during a sample outbound run in a light-off trial. At each time point the color scale represents the decoded probability for every position (normalized to the sum of probabilities across positions) and the red line shows the animal's actual position over time. **(F)** The average decoding error in light-off trials of each session (represented by a dot) decreases with increasing number of simultaneously recorded cells. Dashed line indicates the threshold of >10 simultaneously recorded cells for a session to be included in subsequent analysis. **(G–H)** Decoding error did not increase **(G)** and the percentage of correctly decoded positions did not decrease **(H)** during light-on trials in either the sample (top; $n$ = 15, 16 and 8 sessions with 19.9 ± 1.5, 20.2 ± 2.0 and 15.8 ± 1.1 simultaneously recorded cells) for ArchT-dHPC, ArchT-vHPC and GFP groups, respectively) or the choice (bottom; $n$ = 15, 13 and 11 sessions with 20.3 ± 1.4, 20.8 ± 1.9 and 14.8 ± 1.0 simultaneously recorded cells) phase in any of the experimental groups. Each line represents the decoding results in one session. Error bars indicate mean ± s.e.m. across sessions. The data underlying this figure can be found at https://doi.org/10.12751/g-node.ls2xxj.

light-off trials in the choice phase than for the corresponding trials in the preceding sample phase (group × phase ANOVA; main effect of group: $p < 0.05$; main effect of phase: $p < 0.01$; interaction: $p < 0.05$; ArchT-dHPC mice: $p < 0.01$; ArchT-vHPC mice: $p = 0.81$, GFP mice: $p = 0.86$, Wilcoxon signed-rank test). Similar results were obtained when analysis was restricted to pPYRs, but not pINTs (S6C Fig). To verify that the stability of spatial firing patterns was not generally different

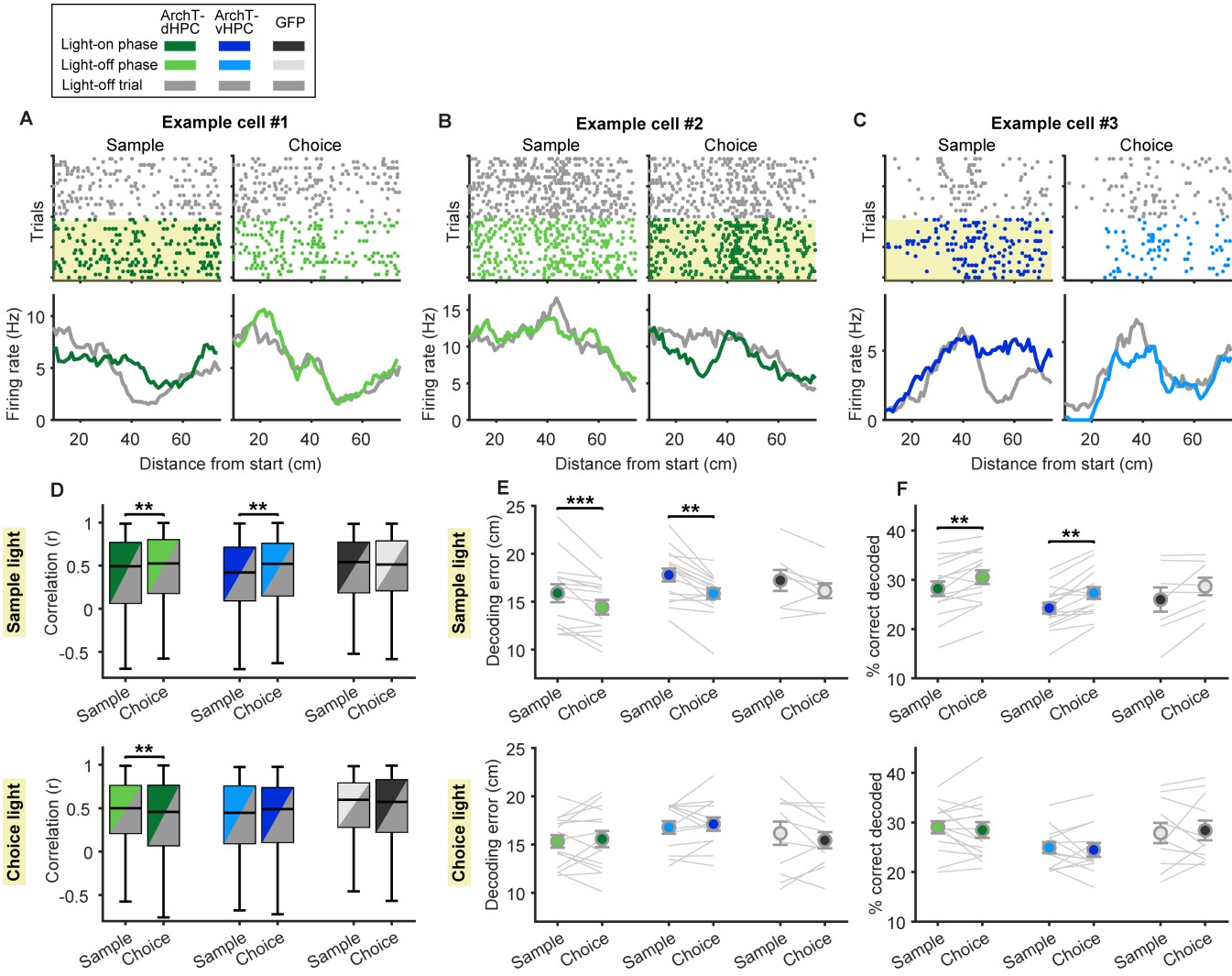

**Fig 4. Spatial firing patterns of prefrontal neurons are altered by dHPC or vHPC silencing in specific SWM phases. (A–C)** Raster plots (top) and averaged linearized firing rates (bottom) of example neurons in the sample (left) and choice phase (right) show changes in spatial firing patterns when the dHPC is inhibited in the sample **(A)** or choice phase **(B)**, or when the vHPC is inhibited in the sample phase **(C)**. Yellow shaded area indicates the trials and task phase in which light was delivered. Gray indicates activity in light-off trials; darker colors indicate activity in light-on trials during the task phase in which light was delivered (light-on phase); light colors indicate activity in the phase where light was not delivered during light-on trials (light-off phase). **(D)** Linearized firing rates of prefrontal neurons (*n* values as in Fig 3C) were correlated between light-off and light-on trials, separately for the task phase in which light was (dark colors) or was not (light colors) delivered. This revealed significant changes in spatial firing patterns when the dHPC is inhibited in the sample (top) or choice phase (bottom), or selectively when the vHPC is inhibited in the sample phase (top). Box plots represent the median (line), 25th and 75th percentiles (box) and the whiskers extend to the minimum and maximum values within 1.5 times the interquartile range below and above the 25th and 75th percentiles, respectively. **(E–F)** A Bayesian decoding algorithm was trained to decode animals' position based on the linearized firing rates of neuronal populations in light-off trials. Using the same algorithm to decode the animal's position during the respective light-on trials resulted in a higher decoding error **(E)** and a lower percentage of correctly decoded positions **(F)** when the dHPC or vHPC was inhibited in the sample phase (top), but not when they were inhibited in the choice phase (bottom). Each line represents the decoding results in one session (*n* values as in Fig 3G–3H). Error bars indicate mean ± s.e.m. across sessions. **p < 0.01 and ***p < 0.001, Wilcoxon sign rank test. The data underlying this figure can be found at https://doi.org/10.12751/g-node.ls2xxj.

in sample and choice phase, we correlated linearized firing rates of even and odd light-off trials separately within the sample and choice phases. A two-way ANOVA (group × phase) showed no main effect of phase ($p = 0.31$) and no interaction ($p = 0.90$). We also found that the strength of the correlation between even and odd light-off trials did not differ between the sample and the choice phase in individual groups ($p > 0.07$, Wilcoxon signed-rank test; S7A Fig). The effects of dHPC and vHPC silencing on spatial firing patterns of PFC neurons were also confirmed for each phase by comparing the similarity of linearized firing rates within light-off and light-on trials to their similarity across these two trial types (S7B Fig).

To examine the alteration of spatial coding caused by hippocampal silencing at the population level, we used the same Bayesian decoding approach described above (Fig 3E–3H) but this time trained the decoder using light-off trials and tested it on light-on trials. As with the correlational analysis, the performance of the decoder was quantified in the task phase in which hippocampal activity was silenced and compared with its performance in the task phase without hippocampal silencing (see section "Methods"). Consistent with the findings from the single-neuron analyses, this resulted in decreased decoding performance when either the dHPC or the vHPC were silenced during the sample phase, measured either using decoding error (Fig 4E) or the percentage of correctly decoded positions (Fig 4F). A two-way ANOVA (group × phase) on either of these measures revealed a significant main effect of phase ($p < 0.05$), but no effect of group and no interaction ($p > 0.15$). Silencing of either dHPC and vHPC caused an increase in decoding error ($p < 0.001$ for ArchT-dHPC mice, $n = 15$ sessions; $p < 0.01$ for ArchT-vHPC mice, $n = 16$ sessions; Wilcoxon signed-rank test) and a decrease in the percentage of correctly decoded positions ($p < 0.01$ for ArchT-dHPC and ArchT-vHPC mice) whereas light delivery in GFP mice had no effect on these measures ($p > 0.10$, $n = 8$ sessions). In contrast, dHPC or vHPC silencing during the choice phase did not affect decoding performance. A two-way ANOVA (group × phase) on either decoding measure showed no effect of group ($p > 0.64$) or phase ($p > 0.29$) and no interactions ($p > 0.25$). This is in contrast to the results from individual neurons, where silencing of the dHPC altered spatial firing patterns in the choice phase (Fig 4D). Taken together, these results suggest that silencing of dHPC and vHPC alters the coding of relative position by PFC neurons in a phase- and subregion-dependent manner, both at the level of individual neurons and neuronal populations.

## The ventral but not the dorsal hippocampus is required for encoding the goal location in prefrontal neurons

To successfully solve the SWM task and obtain available rewards, it is critical for the animal to encode and remember which of the two goal arms was visited in the sample phase. Recent studies could show that prefrontal neurons modulate their firing rate with respect to the goal location [34,49,54–57] and that direct input from the vHPC is required selectively for encoding the goal location in the sample phase but not the choice phase [39]. However, the contribution of the dHPC to goal encoding by PFC neurons is not known. We therefore investigated whether silencing of the dHPC also affected PFC goal representations. To this end, we calculated the firing rate of each neuron for left and right trajectories when the animal approached the goal (outbound), when the animal returned from the goal to the start box (inbound), and when the animal was briefly stationary at the goal between outbound and inbound trajectories (Fig 5A). We then identified neurons as goal-selective that fired significantly stronger when the animal was in one goal arm compared to the other (two-way ANOVA of goal and position, $p < 0.05$ for factor goal, see section "Methods"). Two example neurons displaying goal-selective firing are shown in Fig 5B–5C. For each goal-selective neuron, we computed a goal index for every linearized position in the T-maze (see section "Methods"), representing the difference in firing rate between the neuron's preferred and nonpreferred goal. In accordance with previous findings [39], we found that silencing the vHPC reduced goal-selective firing in prefrontal neurons specifically when the animal had to encode the goal location in the sample phase (Fig 5B and 5E; two-way ANOVA for light and position: significant effect of light, $p < 0.01$, significant effect of position, $p < 0.0001$, and no interactions, $p = 0.25$; $n = 91$), but had no effect on goal representations in the choice phase (Fig 5E; two-way ANOVA with significant effect of position, $p < 0.0001$, no effect of light and no interaction, $p > 0.79$; $n = 76$). In contrast, silencing of the dHPC did not impair goal-selective firing in any of the task phases (Fig 5D); that is, the goal index across positions was not reduced in light-on compared to light-off trials (two-way ANOVA with significant effect of position, $p < 0.0001$, but

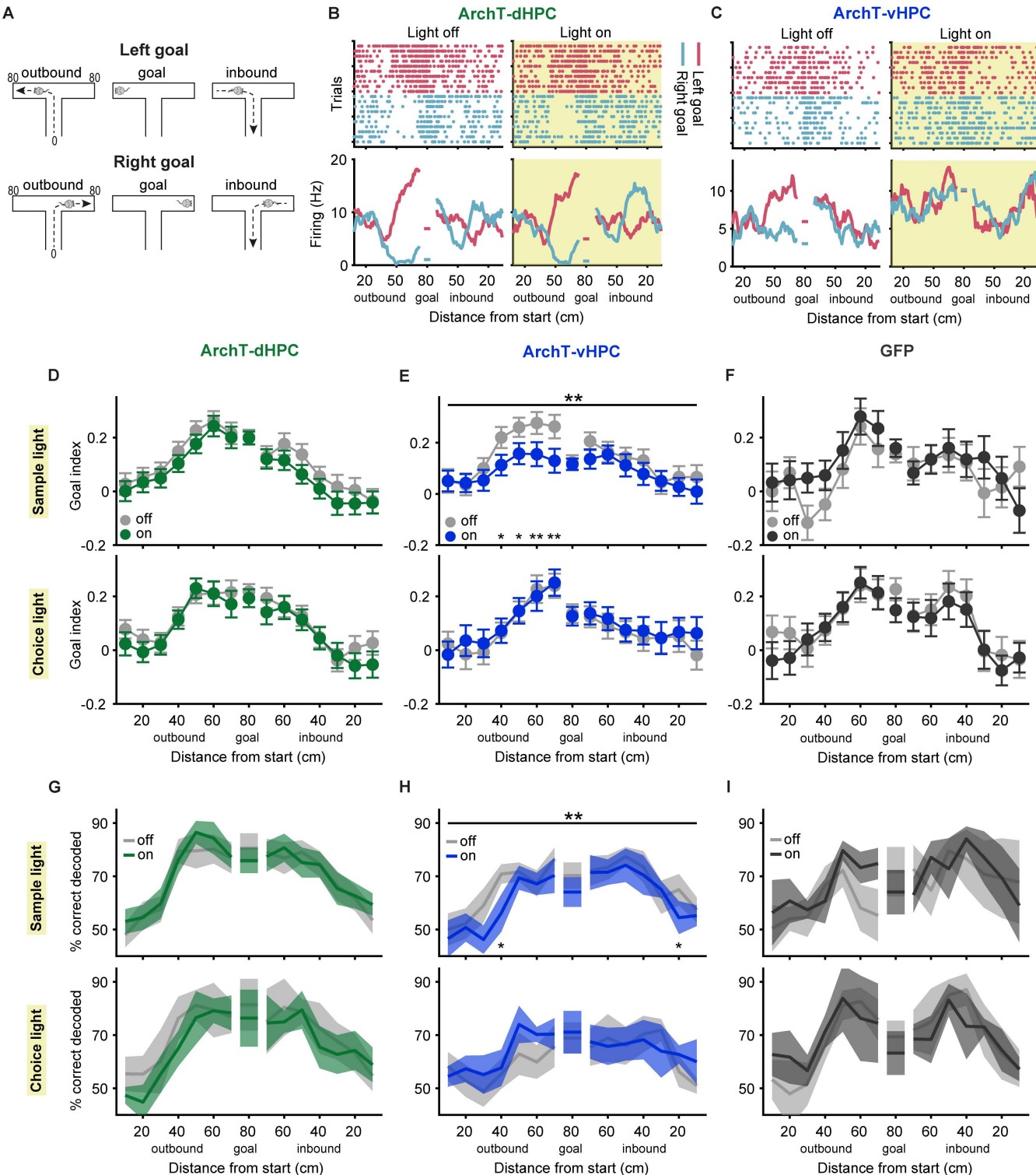

**Fig 5. Prefrontal goal coding is perturbed selectively by vHPC silencing in the sample phase. (A)** Schematic depicting the animal's trajectory approaching the left or right goal (outbound), sitting at the goal and returning from the goal (inbound). Distance from the start is indicated for both goals on the left in cm. **(B)** Raster plot (top) and averaged firing rate (bottom) of an example neuron displaying goal-selectivity in the sample phase of light-off trials (left) by increasing its firing rate when the animal approaches the left goal (red). Firing rate is shown as a function of linearized position for outbound and inbound trajectories and when the animal sits at the goal, as depicted in **A**. During dHPC inhibition in the sample phase (right, yellow

shaded area), this neuron still shows stable goal coding. **(C)** Raster plot (top) and averaged firing rate (bottom) of an example neuron as described in **B**. In trials in which the vHPC is inhibited in the sample phase (right, yellow shaded area), this neuron does not discriminate between the two goals. **(D–F)** Goal index representing the difference in firing rate between preferred and nonpreferred goals in the sample (top; $n$ = 133, 91 and 52 neurons for ArchT-dHPC, ArchT-vHPC and GFP groups, respectively) or choice (bottom; $n$ = 130, 76 and 60 neurons) phase of light-off and light-on trials as a function of position, as depicted in **A**. vHPC inhibition in the sample phase reduces goal selectivity (**E**, top). Upper asterisks indicate $p$ values for an effect of light in a two-way ANOVA (position × light); lower asterisks indicate $p$ values for sign-rank tests comparing light-off and light-on trials at individual position bins; *$p < 0.05$, **$p < 0.01$. Error bars and shaded areas indicate mean ± s.e.m. across neurons. **(G–I)** Decoding of the visited goal arm in the sample (top; $n$ = 15, 12 and 6 sessions with 23.7 ± 1.7, 27.1 ± 2.6 and 19.3 ± 0.7 simultaneously recorded neurons for ArchT-dHPC, ArchT-vHPC and GFP groups, respectively) or choice phase (bottom; $n$ = 6, 7 and 8 sessions with 22.7 ± 1.9, 28.0 ± 3.6 and 20.8 ± 1.1 simultaneously recorded neurons) across positions, as indicated in **A**, using a support vector machine classifier trained and tested on neuronal population activity during light-off or light-on trials. Decoding accuracy is reduced by vHPC inhibition in the sample phase (**H**, top). Upper asterisks indicate $p$ values for an effect of light in a two-way ANOVA (position × light); lower asterisks indicate $p$ values for sign-rank tests comparing light-off and light-on trials at individual position bins; *$p < 0.05$, **$p < 0.01$. Error bars and shaded areas indicate mean ± s.e.m. across sessions. The data underlying this figure can be found at https://doi.org/10.12751/g-node.ls2xxj.

no effect of light and no interactions, $p > 0.15$; $n$ = 133 for ArchT-dHPC sample light, $n$ = 130 for ArchT-dHPC choice light). Similarly, light delivery in GFP mice did not affect prefrontal goal representations (Fig 5F; significant effect of position, $p < 0.001$, no effect of light and no interactions, $p > 0.24$; $n$ = 52 for GFP sample light and $n$ = 60 for GFP choice light).

To examine goal-selective firing at the populational level, we trained a support vector machine classifier to decode the visited goal arm using firing rates of simultaneously recorded neurons in each session (see section "Methods"). In light-off trials, goal identity could be decoded with above chance accuracy for all positions in the goal arm and in the stem during the inbound trajectory for both sample and choice phases (Fig 5G–5I; $p < 0.0001$, $z$-test with Bonferroni correction for multiple comparisons). When we trained and tested the classifier on light-on trials, we found that silencing the vHPC in the sample phase reduced decoding accuracy compared to light-off trials (Fig 5H; two-way ANOVA for light and position with a significant effect of light, $p < 0.01$, a significant effect of position, $p < 0.0001$, and no interaction, $p = 0.65$; $n$ = 12 sessions). In contrast, dHPC silencing in the sample phase did not reduce decoding accuracy (Fig 5G; two-way ANOVA with significant effect of position, $p < 0.0001$, but no effect of light and no interaction, $p > 0.46$; $n$ = 15 sessions), similar to sample light delivery in GFP mice (Fig 5I; significant effect of position, $p < 0.001$, no effect of light and no interaction, $p > 0.09$; $n$ = 6 sessions). In the choice phase, neither silencing of the ventral nor of the dorsal HPC had any effect on goal decoding (Fig 5G–5I; significant effect of position, $p < 0.0001$, no effect of light and no interactions, $p > 0.27$; $n$ = 6 sessions for ArchT-dHPC and $n$ = 7 sessions for ArchT-vHPC). Taken together, these results confirm the critical role of the vHPC in prefrontal goal encoding, but also demonstrate that the dHPC, in contrast, does not contribute to these goal representations in the prefrontal cortex. We did find, however, that dHPC silencing in the sample phase, but not the choice phase, impaired phase-locking of PFC neurons to local theta oscillations, specifically in the goal arms of the T-maze (S8A-D Fig). In contrast, theta phase-locking was not affected by vHPC silencing (S8E Fig). These results indicate that the dHPC does influence PFC activity in a manner that may be relevant for goal encoding, and further highlight the differential effects of dHPC and vHPC silencing on PFC activity during the sample phase.

## Discussion

Lesion and pharmacological inactivation studies have shown that both the dHPC and vHPC are required for SWM [17,18,58]. However, given that SWM tasks consist of several phases with different cognitive demands, it has remained unclear to which phases the dHPC and vHPC contribute. Our results reveal that the two hippocampal poles play both distinct and complementary roles in SWM. First, we found that activity in both dHPC and vHPC was necessary during the 'sample' phase of the task, where to-be-remembered information is encoded. A previous study found that vHPC inputs to PFC were also required only during the sample phase [39]. Our results confirm and extend this finding to the entire vHPC. Although the dHPC contributes to the encoding of spatial information during the sample phase, our electrophysiological

results suggest that the two HPC subregions make distinct contributions to this process (discussed further below). Second, we found that only the dHPC was required during the choice phase of the task, where the remembered information needs to be used to guide an upcoming decision. In line with this finding, the activity of dHPC pyramidal neurons has been shown to reflect not only the animal's current position but also its upcoming choices [57,59–61]. Interestingly, a recent study found that activity in the left, but not the right, CA3 region of dHPC was selectively required during the choice phase of SWM, suggesting a lateralization of the functional contributions of dHPC during SWM [62]. In contrast to their involvement in the sample and choice phases, neither the dHPC or the vHPC were required during the delay phase, where the to-be-remembered information needs to be maintained. Consistent with this finding, information about previously visited locations is not strongly represented in the activity of dHPC neurons during the delay period of SWM tasks [57,63,64]. Interestingly, this information is also not represented by PFC neurons [39,49,54,65], although PFC activity during the delay period is necessary for SWM performance [49,54]. In which brain region(s) the information about previously visited locations is maintained in SWM is therefore an open question for future research. Alternatively, it is possible that information is not maintained during the delay period through the sustained activity of PFC or HPC neurons but rather via 'activity-silent' mechanisms such as short-term changes in the pattern of their synaptic weights [66,67]. Such patterns could support the generation of goal-specific activity patterns at a later time point, allowing the retrieval of the remembered information during the choice phase.

A modest decrease in the animals' running speed could also be observed during dHPC and vHPC silencing. It is unlikely that the speed decrease reflects a general state of uncertainty associated with behavioral errors or a motor deficit that indirectly contributed to such errors. Notably, running speed was decreased during all silencing conditions, including those which had no effect on performance (i.e., silencing of vHPC during the choice phase). Furthermore, silencing of either subregion in the sample phase did not affect running speed in the choice phase, despite causing impaired performance. Indeed, our covariance analysis suggests that the impairments in SWM performance and the decrease in running speed represent independent consequences of hippocampal silencing. Although it remains unclear how HPC silencing could influence running speed, one possibility is that this could be mediated by projections from the hippocampus to the lateral septum [68].

A number of studies suggest that the HPC supports SWM in part by interacting with, and influencing activity in, the PFC [17,18,30–32,39,61]. Consistent with such an influence, we found that silencing either the dHPC or the vHPC altered the spontaneous firing rates of PFC neurons. The vHPC has long been known to send direct monosynaptic inputs to the PFC [50,69–71], whereas the dHPC has been thought to project to the PFC primarily via polysynaptic projections involving the nucleus reuniens and rhinal cortices [27,28]. However, more recent studies suggest that the dHPC also projects to the PFC [24,25,43,72], although these projections appear to be considerably weaker than those from the vHPC. Although we found that the effects of silencing the dHPC and vHPC on the spontaneous activity of PFC neurons were similar in magnitude, vHPC silencing elicited changes at a shorter latency, possibly reflecting the stronger direct projections to the PFC from the vHPC.

The HPC is critical for representing animals' position in space and enabling spatial learning and behavior [1–3]. It is likely that spatial information is relayed from the HPC to its downstream targets, where it is integrated with other sources of information and used for guiding behavior. The PFC is one of the main cortical projection targets of the HPC and HPC-PFC interactions in the form of synchronized neuronal activity between the two structures have been widely reported [29–32,34,40]. These interactions likely reflect, at least in part, the influence of the HPC on the PFC [29,31,34]. Based on these findings we hypothesized that hippocampal silencing would impair spatial coding in the PFC. Unexpectedly, however, neither silencing of the dHPC nor the vHPC impaired the encoding of the animals' relative position along the T-maze by PFC neurons. Silencing did, however, alter their specific spatial firing patterns, that is at which locations the neurons were more or less active. In other words, the spatial code rather than spatial coding ability per se, was altered by hippocampal silencing. This suggests that the hippocampus may be only one of several inputs that relay spatial information to

the PFC. The exact spatial firing patterns of PFC neurons may reflect integration of information from these inputs, in which case interrupting any one of them will alter those firing patterns, as we found. Candidate input regions other than the hippocampus include the lateral entorhinal and perirhinal cortices, which project directly to the PFC [73] and whose neurons are spatially modulated [74]. It should also be emphasized that the role of the hippocampus is not restricted to spatial processing, which may be only one manifestation of a more general cognitive function [2,75]. HPC-PFC interactions are also observed during non-spatial tasks [19], raising the possibility that the influence of the HPC may manifest itself in other aspects of PFC activity beyond spatial processing.

A correct decision in the choice phase of our SWM task requires the animals to remember which goal arm they visited during the sample phase. This information is encoded by PFC neurons, many of which show differential activity depending on whether the animals enter the left or right goal arm during the sample phase [39,49,56]. We found that silencing the vHPC during the sample phase impaired such goal arm encoding, which could explain why this manipulation also impaired task performance. These results are consistent with previous studies which have silenced the axon terminals of ventral hippocampal neurons in PFC [39,54]. Silencing these terminals also disrupts anxiety-related spatial firing patterns [76], suggesting that vHPC inputs might support task-related spatial representations more generally. Interestingly, although PFC neurons show the same goal arm preference (i.e., which arm they are more active in) during the sample and choice phases [49], silencing the vHPC during the choice phase did not affect goal arm encoding in the same phase (see also [39]). Thus, goal arm encoding in the PFC only depends on inputs from the vHPC when the goal location has to be remembered, but does not require the vHPC when the goal information is not required for subsequent performance. In contrast to vHPC silencing, goal encoding during the sample phase was unaffected by dHPC silencing. However, encoding of relative position in the T-maze during the choice phase was impaired by dHPC but not vHPC silencing, which could explain why animals made more errors during the choice phase only when the dHPC was silenced.

The dHPC and vHPC have long been viewed as serving distinct functions. The two subregions differ in terms of the genes they express, which brain areas they are connected to, how their neurons represent animals' position and how their perturbation affects behavior (reviewed in [7,8]). Based on these findings, it has been suggested that spatial learning and navigation primarily involve the dHPC rather than the vHPC [8,77]. Supporting this view, lesions of the dHPC consistently impair the learning of spatial locations and the acquisition of spatial tasks whereas vHPC lesions have little or no effect [78,79]. Interestingly however, if the vHPC is lesioned or inactivated following learning, impairments are observed in the retrieval of spatial information [15,80] and in performance on spatial navigation tasks, including SWM [17,18]. This suggests that although the dHPC alone can support spatial learning, both hippocampal subregions are involved in spatial learning and memory in the intact brain. Consistent with this, we found that both dHPC and vHPC were necessary for SWM performance, as has been observed in previous studies [17,18,58]. However, we also found that the two subregions make partly distinct contributions to this process. Our results therefore support the alternative view that spatial processing involves both the vHPC and dHPC, albeit in different ways [5,9–11,13]. This view is further supported by the differential consequences of dHPC and vHPC silencing on spatial representations in the PFC that we observed. Taken together, our results thus further our understanding of the differential contributions made by the dorsal and ventral HPC poles to spatial behavior and information processing.

## Methods

### Subjects

In total, 30 male C57BL/6N mice (Charles River Laboratories, Wilmington, USA) were used in this study, aged 11–12 weeks at the start of the experiments. Fifteen of these mice were used for dHPC silencing, of which 8 were also used to record neuronal data in the PFC. Eight mice were used for vHPC silencing and simultaneous neuronal recording in the PFC. The control group consisted of 7 mice expressing GFP either in the dHPC ($n = 4$) or vHPC ($n = 3$) but were

analyzed together. PFC neuronal activity was recorded in 5 of them. All mice were housed in transparent acrylic cages (35 × 14 × 13 cm) in a ventilated animal scantainer (Scanbur Technologies, Karlslunde, Denmark) at 20–25 °C and 40–60% humidity on a 12 h light/dark cycle. Experiments were performed during the light period. Animals were first housed in groups up until surgical implantation, and were then housed in individual cages for the duration of the experiment. All procedures were approved by the local animal care committee (TVA FU-1038 and FU-1256, Regierungspräsidium Darmstadt, Germany).

## Surgical procedures

Preceding all surgeries, mice were anesthetized in a chamber filled with 4% isoflurane in oxygen, placed in a stereotaxic frame and injected with carprofen (4 mg/kg, subcutaneously) and dexamethasone (2 mg/kg, subcutaneously) for reducing pain and inflammation, atropine (50 µL, intraperitoneal) to decrease mucus secretions and Ringer's solution (0.8 mL, subcutaneously) as fluid replacement. During the surgery, anesthesia was maintained with an isoflurane concentration of 1–2% (in oxygen at a flow rate of 0.35 L/min), which was regularly adjusted based on the monitored breathing rate. Body temperature was maintained at 37 °C with a heating blanket placed under the animal.

For viral infusions, the skull was exposed, and small craniotomies were made above the dHPC or vHPC. An adeno-associated virus (AAV) containing the genetic construct either for the inhibitory opsin ArchT coupled with GFP (rAAV5-CamKII-ArchT-GFP, $5.2 \times 10^{12}$ virus molecules/ml or $7.5 \times 10^{12}$ virus molecules/ml, UNC Vector Core, Chapel Hill, USA) or only GFP (rAAV5-CamKII-GFP, $5.3 \times 10^{12}$ virus molecules/ml, UNC Vector Core, Chapel Hill, USA) was bilaterally infused using a 35-gauge needle attached to a syringe (NanoFil, 10 mL, World Precision Instruments) and a microsyringe pump controller (UltraMicroPump III, World Precision Instruments). For dHPC injections, 1 µL was infused at −2.0 mm AP, ±1.5 ML and −1.40 DV at a speed of 100 nL/min. For vHPC, 3 injections of 300 nL per infusion site (total of 900 nL per hemisphere) and a speed of 50 nL/min were made at (1) −3.15 mm AP, ±3.40 mm ML, −3.00 mm DV, (2) −3.15 mm AP, ±3.40 mm ML, −1.65 mm DV, and (3) −3.15 mm AP, ±3.65 mm ML, −2.10 mm DV. AP and ML coordinates were measured relative to bregma, whereas the DV coordinate was measured relative to the brain surface. After the infusion, the syringe was left in place for an additional 10 min to allow diffusion from the tip. After slowly retracting the needle from the brain, the scalp was sutured closed using a medical sowing kit and lidocaine was applied to the stitches to reduce potential post-surgical pain.

For the implantation of optic fibers and electrode microdrives, animals were anesthetized and placed in the stereotactic frame as described before. First, small craniotomies were drilled into the skull above HPC and PFC. Then, two small screws (0.96 mm, MF-5182, BaSi) were inserted into the skull over the frontal cortex and cerebellum to serve as reference and ground, respectively. Two additional screws were implanted to provide further anchoring support and all screws were secured using ultraviolet glue and dental cement. Optic fibers (#CFML12L10 or #CFML22L20, Thorlabs, 200 µm diameter core, 0.39 NA) were inserted bilaterally into the brain tissue to either target the dHPC (−2.0 mm AP, ±1.5 mm ML, −0.9 mm DV) or the vHPC (−3.15 mm AP, ±3.65 mm ML, −1.60 mm DV). Both optic fibers were fixed to the bone and covered using dental cement mixed with black acrylic paint to reduce light visibility during light delivery.

To obtain neuronal spiking data, a moveable bundle of 8 or 16 stereotrodes was used. The stereotrodes were made made from tungsten wire (0.0005" diameter, California Fine Wire Company, Grover Beach, USA) and were connected to an electrode interface board (EIB; EIB-16, Neuralynx, Bozeman, USA or EIB-32, Open Ephys or ZifClip EIB ZCA-EIB32, Tucker Davis Technologies, Alachua, USA). The stereotrode bundle was attached to a shuttle on a microdrive that could be advanced via a shuttle on a screw base (dDrive-m, NeuroNexus Technologies, Ann Arbor, USA). The whole structure was enclosed with a 3D-printed cover. Before implantation, the stereotrode bundle was coated with a thin layer of fluorescent dye (DiI, ThermoFisher Scientific, Waltham, USA) for later histological verification of placements, and was then inserted into the brain tissue above the left PFC. Stereotrodes were advanced until the tips reached the prelimbic cortex (+1.85 mm AP, −0.35 mm ML, −1.3 mm DV). In 4 animals, a movable 16-channel silicon probe

(A1X16-5 mm-25–177-H16_21 mm mounted on dDrive-m, NeuroNexus Technologies, Ann Arbor, USA) was implanted in the PFC instead of the stereotrode microdrive. After inserting the electrodes into the brain, the craniotomy was covered with Vaseline to ensure lasting mobility of the electrodes. In some animals, an additional tungsten electrode (0.003" diameter, California Fine Wire Company, USA) was implanted in the left dHPC (−2.0 mm AP, −1.5 mm ML, −1.4 mm DV) or vHPC (−3.15 mm AP, −3.65 mm ML, −2.10 mm DV) to record local field potentials. After connecting ground and reference screws to the EIB, the implant was fixed on the skull with dental cement. After the surgery, mice were given at least one week of recovery before the next steps of the experiment.

## Behavioral training and testing with optogenetic silencing

After surgical recovery, the animals' food intake was restricted until they reached 85% of their ad libitum body weight. Mice were then trained on a delayed non-match-to-sample SWM task on an automated T-maze as previously described [49]. The T-maze was elevated 30 cm from a wooden platform and consisted of a start box (16 × 13 cm surrounded by 12 cm high walls), a central arm (stem, 40 cm long, 4.5 cm wide, 4 cm high walls on either side) and two goal arms (36 cm long, 4.5 cm wide, 4 cm high walls on either side). Both goal arms as well as the start box could be closed via 3-D printed rotating doors powered by servomotors (Segelwinden-Servo RS-22 YMB, Modelcraft ). Reward ports were located at the end of each goal arm and in the start box and equipped with an infrared sensor: Upon a nose poke by the animal, a drop of sweetened condensed milk (5–10 μL, Milchmädchen, Nestlé; diluted in water 1:3) was delivered through a cannula using either solenoid valves (Miniature Inert Liquid Valve, Parker Hannifin GmbH, Bielefeld, Germany) or peristaltic pumps (12V dosing pump, Grothen). Three additional infrared sensors were attached at the entrance of the two goal arms and at the entrance of the stem after the start box to detect the animal's position. A crossing of the infrared beams was detected by an Arduino microcontroller (Arduino Mega, Arduino LLC, Somerville, USA). Based on the animal's position (i.e., which infrared beam was crossed) and the previously defined trial sequence, the microcontroller triggered door opening and closing as well as reward delivery. Throughout all behavioral sessions, the recording chamber was dimly lit by an array of LEDs hanging from the ceiling, and every session was video recorded.

Each SWM trial consisted of three phases, the sample phase, the delay phase, and the choice phase (Fig 1C). Before each trial, the animal was confined to the start box. At the beginning of the sample phase, the start box door opened, and the animal was able to move through the maze to receive a reward at the end of one goal arm (sample goal), which was randomly chosen for every trial. Access to the opposite goal arm was blocked by a door. The mouse then returned to the start box, where another reward was delivered and the closing of the start box door marked the end of the sample phase. The mouse was now confined to the start box for a delay period of 15 s. In the following choice phase, the animal could enter both goal arms, but reward was only delivered at the end of the arm opposite to the sample goal. This was denoted as a correct choice. Entry into either of the goal arms triggered the closing of the door in the opposite arm to prevent a late change of decision. In case of a correct choice, the animal also received a reward in the start box after return, but this reward was omitted following an incorrect choice. The mouse then remained in the start box for an inter-trial-interval (ITI) of 40 s.

Training began with 2 days of habituation to the T-maze, followed by 2 days of shaping with 10 trials per session, in which the animals were forced to always make a correct choice by blocking access to the incorrect choice goal arm. Thereafter, animals received daily training sessions, in which they performed 10 trials with both goal arms open in the choice phase and therefore could make mistakes. Criterion performance was reached once the mice made at least 70% correct choices on three consecutive training days. In the following testing sessions, animals performed 40 trials per session with light delivered for optogenetic silencing in 50% of the trials, which were chosen pseudorandomly to ensure the same number of trials with left and right sample goals. Light delivery was temporally restricted to one of the 3 task phases. In the sample phase, light was delivered continuously from start door opening until reward delivery in the start box after return from the goal arm. In the delay phase, light was delivered for the full duration of the delay. In the choice phase, light

was delivered from the opening of the start door until the mouse had reached the goal and initiated its return by leaving the goal area. Yellow light was delivered from a DPSS 594 nm laser (Omicron Laserage, Rodgau-Dudenhofen, Germany) via a dual patch cord (200 μm core diameter, 0.22 NA, Doric Lenses, Quebec, Canada) attached to a rotary joint (Doric Lenses, Quebec, Canada) to the optic fibers implanted in the HPC. Laser intensity was set to 8 mW for dHPC inhibition and to 16 mW for vHPC inhibition to account for the size differences of the areas. Light on- and offsets were automatically triggered by an Arduino microcontroller (Arduino Mega, Arduino LLC, Somerville, USA).

### Electrophysiological recording during optogenetic manipulations

During the SWM task, neuronal activity in the PFC was recorded, amplified and digitized using a 32-channel Neuralynx recording system (Neuralynx Digital Lynx, Neuralynx, Bozeman, USA) and a headstage (HS-18, Neuralynx, Bozeman, USA; or 16- or 32-Channel ZIF-Clip Headstage, ZC16 or ZC32, Tucker Davis Technologies, Alachua, USA) connected to the implanted microdrive. Signals were recorded broadband with a bandpass filter of 1–6,000 Hz and a sampling rate of 32 kHz referenced against the animals' reference screw. The animal's position in the maze was registered by tracking a red LED connected to the headstage with a video tracking camera (Neuralynx, Bozeman, USA) sampled at 25 Hz. Behaviorally relevant events throughout the task were delivered from the Arduino microcontroller to the recording system as TTL pulses.

Additionally, neuronal responses to optogenetic perturbation were tested in approximately 10 min long sessions, during which the animal moved freely in a small open field box (30 × 25 × 30 cm) while brief light pulses were delivered to the dHPC and vHPC (pulse duration of 500 ms or 1 s and an ITI of 3 s). Neuronal activity was recorded throughout the session as described before. These sessions (termed as "brief light sessions") were used to assess the effect of hippocampal inhibition on neuronal activity in the PFC while the animal was freely moving and not performing any behavioral task.

### Perfusion and histology

Animals with implanted electrodes received small brain lesions before perfusion to identify electrode locations. For this, mice were briefly anesthetized in a chamber filled with approximately 4% isoflurane in oxygen and a 50 μA current was applied for 9 s to 2 electrodes in the PFC and to the electrode in the HPC. For perfusion, mice received an intraperitoneal injection of sodium pentobarbital (0.3 ml, Narcoren, Merial GmbH, Hallbergmoos, Germany). Once pedal reflexes were absent, animals were transcardially perfused with 4% paraformaldehyde, 15% picric acid in PBS (PFA). Brains were extracted and fixated in PFA solution overnight before transfer to storing solution (50 g of sucrose and 0.25 g of sodium azide in 500 ml of phosphate-buffered saline, PBS). For histology, brains were cut on a vibratome (Leica VT1000S, Leica Biosystems, Nussloch, Germany) into coronal sections at a thickness of 80 μm. To enhance the visibility of GFP expression, brain sections were incubated overnight with a primary GFP antibody (rabbit anti-GFP, 1:1000, ThermoFisher Scientific, Waltham, USA) in carrier solution (1% horse serum, 0.5% Triton X-100% and 0.2% BSA in PBS) followed by overnight incubation with a secondary green fluorescent antibody (anti-rabbit 488 Alexa Fluor, 1:750, ThermoFisher Scientific, Waltham, USA) in carrier solution. To visualize cell bodies, the sections were then incubated in a DAPI solution (1 mg/ml in PBS; ThermoFisher Scientific, Waltham, USA) for 5 min. Brain sections were mounted on microscope slides to verify electrode placements and virus expression with a confocal microscope (Eclipse 90i, Nikon, Minato, Japan).

### Spike detection and clustering

For analyzing neuronal firing rates, the raw electrophysiological signals were first preprocessed by subtracting from each electrode signal the median signal across all electrodes at every time point, as a common reference [81]. To extract spiking activity, the signal was high pass filtered for ≥300 Hz and a third order Butterworth filter was applied. Then, spikes were detected using a double-threshold flood fill algorithm (SpikeDetekt) and sorted into single-unit clusters based on

their waveform principal components using the semi-automatic algorithm Klusta [82]. Clustering was followed by manual refinement based on visual inspection of spike waveforms as well as auto- and cross-correlograms of each cluster using phy Kwik GUI. Finally, every cluster was further analyzed in Matlab and several quality metrics, such as its signal-to-noise ratio, inter-spike interval, minimum spike amplitude and average firing rate, were calculated. Units were included in subsequent analyses if they met our established quality criteria (signal-to-noise ratio > 2, <0.5% of all spikes with an inter-spike interval <1 ms, average firing rate ≥ 1 Hz). We also computed for each neuron its average spike waveform from the channel on which the waveform was largest. The half spike width (valley width at half minimum) and valley-to-peak separation of each neurons' waveform were then used to separate neurons into putative pyramidal neurons and interneurons. To this end, the distributions of these two waveform features were fit using a 2-dimensional Gaussian mixture model [83]. Neurons with low classification confidence ($p < 0.95$ of belonging to the assigned class) were excluded from analyses comparing putative pyramidal neurons and interneurons.

## Data analysis

### Behavioral analysis during SWM

For the analysis of behavior, mice were separated in three groups: Mice belonging to the ArchT-dHPC group expressed the inhibitory opsin ArchT in dHPC pyramidal neurons, mice in the ArchT-vHPC group expressed the opsin in vHPC. The control group contained mice that expressed GFP, but no opsin, in either dHPC or vHPC. Since performance was similar in mice expressing GFP in the dHPC and mice expressing GFP in the vHPC, they were combined to one group for subsequent analysis (GFP group). SWM performance for each animal was assessed by calculating the percentage of correct trials in light-off and light-on trials across sessions with light delivery in the same task phase (light on in sample phase, light on in delay phase or light on in choice phase, 2–4 sessions of each type per animal).

To analyze the animal's traveled distance in each trial between the starting point and the goal of the maze, the animal's two-dimensional position obtained by the electrophysiological recording system was converted and normalized to represent the animal's position in cm relative to a starting point, which was defined as the beginning of the center arm. To this end, we created a one-dimensional linearized position vector by calculating for every position its distance from the starting point until the animal reached the goal at the end of the goal arm. This way, linearized position values from 0 to 40 cm represented positions on the stem, values from 41 to 80 cm represented positions in either of the goal arms and values below 0 indicated positions in the start box (Fig 3A). The linearized position was furthermore used to obtain the animal's speed of movement by calculating the position difference between each two behavioral samples and dividing this value by their difference in time. Running speed was then averaged separately for each sample and choice run, from the maze start to the goal. For each session, the median running speed was calculated across all sample and choice runs, separately for light-off and light off trials. For each animal, the median running speed was similarly computed across all runs in all sessions.

### Analysis of neuronal firing rates recorded during brief light sessions

To analyze firing rate modulations in PFC neurons recorded during brief light sessions, in which the animal moved freely in an open field for approximately 10 min and light was repeatedly delivered to the dHPC or vHPC in 0.5- or 1-s pulses (see above), each neuron's firing rate was averaged over a period of 0.5 s following light onset and then compared to a period of 0.5 s immediately before light onset (baseline activity). Neurons were defined as excited or inhibited when their firing rate in this window was significantly higher or lower than their baseline ($p < 0.05$, Wilcoxon signed-rank test). To compare modulated neurons across groups, we created a peri-stimulus time histogram (PSTH) for each neuron ±500 ms around light onset with bins of 10 ms and normalized this firing rate by subtracting the neuron's baseline activity from each bin.

To estimate the latency to excitation or inhibition for each neuron [84], we generated surrogate PSTHs by calculating firing rates in 10 ms bins during the 1 s period before light onset in each trial and shifted the firing rates randomly in time. We then subtracted the first half of each surrogate PSTH from the second half. This process was repeated 1,000 times to calculate a distribution of firing rate modulations that would be expected from random activity. We then found the first time bin when the actual firing rate modulation was higher or lower than 95% of the random distribution for at least 2 consecutive bins and defined this as the response latency. Neurons that did not meet this criterion within the first 200 ms after light onset were excluded from this analysis.

### Analysis of linearized firing rates of neurons recorded during SWM

For each recorded neuron in the PFC, its linearized firing rate was calculated by counting the number of spikes while the animal was within a specific linear position bin and dividing this number by the time the animal spent in the same bin. Bins were only included if the animals' running speed was at least 5 cm/s, in order to limit the analysis to periods of running. Linearized positions were calculated as described above (see section "Behavioral analysis during SWM"). We differentiated between outbound trajectories, which started when the animal left the start box and ended when it reached the goal, and inbound trajectories, which began when the animal left the goal and ended when it returned to the start box. Linearized firing rates were calculated separately for the sample and the choice phase. For the analyses shown in Figs 3 and 4, linearized rates were calculated for left and right trials combined in order to capture generalized coding of the animal's position relative to the goal. To examine goal-specific representations, linearized firing rates were calculated separately for left and right trials for the analyses shown in Fig 5 (see section "Analysis of goal-selective firing", below).

To identify neurons that significantly coded for position, we calculated linearized firing rates for outbound trajectories of every trial using linear position bins of 10 cm and then applied repeated measures ANOVA (Matlab function ranova) with the factor position, separately for light-off and light-on trials in sample and choice phase. Neurons were considered to be significantly modulated by position in either of the trial types when the $p$-value was $<0.05$. We then compared the fraction of modulated neurons between light-on and light-off trials. We note that this criterion for classifying cells as spatially modulated is relatively permissive in that it does not require a certain minimum strength of spatial modulation, as would be observed in hippocampal place cells for example. However, similar results were obtained when neurons were classified based on the strength of their spatial modulation, as quantified using their spatial information (see below).

We calculated each neuron's spatial information, which measures how selectively a neuron fires within an environment. High values indicate that a cell fires only in a small part of the environment but is mostly silent at other positions, whereas values close to zero indicate that a neuron fires with a similar frequency across all positions. We calculated each neuron's linearized firing rate separately for light-off and light-on trials and then applied the equation for spatial information (bits/spike) according to [52]:

$$Spatial\ information\ =\ \sum_{i=1}^{N} p_i \frac{f_i}{f}\ log_2 \frac{f_i}{f}$$

(1)

where $N$ is the total number of spatial bins, $p_i$ is the occupancy probability in the ith bin, $f_i$ is the firing rate in the ith bin and $f$ is the average firing rate of this neuron across all position bins.

To analyze the similarity of spatial firing patterns between light-on and light-off trials, we used spatial bins of 1 cm and smoothed the linearized firing rate with a running average across 10 cm. The smoothed linearized firing rates of outbound trajectories were then averaged separately across sample and choice phases, and across light-off and light-on trials, resulting in 4 linearized firing rates for each neuron (sample light-off, sample light-on, choice light-off and choice light-on). Next, we calculated the Pearson correlation coefficient ($r$) between linearized firing rates of light-on and light-off trials, separately for the sample ($r_{sample}$: sample light-off versus sample light-on) and choice phase ($r_{choice}$: choice light-off versus

choice light-on). Since the optogenetic manipulation took place only in one of the phases within a session, the correlation between light-off and light-on trials in this phase was compared with the same correlation in the other phase, which served as a control. For example, in sessions in which light was delivered in the sample phase, $r_{sample}$ quantified the effect of light delivery on spatial firing patterns whereas $r_{choice}$ served as a control. This analysis assumes that the stability of linearized firing rates is similar in sample and choice phases. To verify this, we correlated each neuron's linearized firing rate between even and odd light-off trials and then compared the correlation coefficients between sample and choice phase for each experimental group (S7A Fig).

## Bayesian decoding of linearized position

We used the population activity of neurons recorded simultaneously in each session to decode the linearized position of the animal in every trial by applying a Bayesian decoding algorithm [85]. To this end, we calculated for every neuron its linearized firing rates using bins of 10 cm, as described above. Linearized firing rates were calculated using all trials except the one in which position was to be decoded. For this trial, we calculated for each neuron its spike counts in overlapping time bins of 125 ms (100 ms overlap between bins) from the beginning to the end of the outbound trajectory of the trial. A maximum likelihood approach was used to estimate the animal's position in every time bin, as described elsewhere [51,53,86]. Briefly, the probability ($P$) of the animal's position (pos) given a certain number of spikes can be calculated applying the Bayesian rule:

$$P(pos \mid spikes) \ = \ \frac{P(spikes \mid pos) \ \times \ P(pos)}{P(spikes)}$$

(2)

We assumed a uniform probability across positions. Additionally, we assumed that neurons fire independently of each other and that their firing rate follows a Poisson distribution. We can therefore apply the Poisson probability density function to calculate the probability of the observed number of spikes in a certain position as:

$$P(spikes \mid pos) \ = \ \prod_{i=1}^{N} \frac{\tau f_i(pos)^{n_i}}{n_i!} \ e^{-\tau f_i(pos)}$$

(3)

where $N$ is the total number of cells recorded in the session, $\tau$ is the length of the time bin, $f_i(pos)$ is the linearized firing rate of the $i$th cell, and $n_i$ is the spike count of the $i$th cell in the current time bin. By normalizing $P(pos \mid spikes)$ to its sum over all possible positions, we can avoid estimating the probability of occurring spikes ($P(spikes)$). The position with the highest probability $P$ was then taken to be the decoded position for the current time bin.

To compare the decoding performance with and without optogenetic manipulations, we first trained the Bayesian algorithm by calculating each neuron's linearized firing rates from sample and choice phases during all light-off trials except one. The remaining trial was then used to test decoding performance in the same phase. This was repeated until every light-off trial was used once for testing. For each session, we then computed the percentage of correctly decoded positions and the mean decoding error by subtracting the animal's real position from the decoded position and taking its absolute value. The same procedure was repeated using light-on trials of every session. To test decoding performance during optogenetic manipulations based on spatial firing patterns under control conditions, we calculated each neuron's linearized firing rates during the sample or choice phase of light-off trials and used them to decode the animal's position in the same phase during each light-on trial. We then compared the mean decoding error and the percentage of correctly decoded positions for sample and choice phase; since the optogenetic manipulation took place only in one of the phases within a session, decoding accuracy values in the respective other phase served as a control. To ensure overall high decoding accuracy, we only included sessions with more than 10 simultaneously recorded neurons in these analyses.

## Analysis of goal-selective firing

To analyze firing rate differences between the two goal arms, we calculated linearized firing rates, as described above, but separately for left and right trajectories, in both outward and inward directions. Linearized firing rates were calculated using 20 cm bins, with a 10 cm overlap between bins. We also calculated firing rates during the period in which the animal was briefly stationary at the end of the goal arm ('goal'), in many cases while consuming the reward. This period was defined as when the animal was moving with a speed of ≤5 cm/s and ≤5 cm from the goal (Fig 5A). We then identified neurons with a significantly higher firing rate in one of the goal arms compared to the other by applying a two-way ANOVA (goal × position) across three position bins for each goal arm: the averaged firing rate in the goal arm during the outbound trajectory (position 40−80), the averaged firing rate while the animal was stationary at the goal, and the averaged firing rate in the goal arm during the inbound trajectory of the goal arm (position 80−40). The ANOVA was performed separately for light-off and light-on trials in the sample or choice phase. Neurons with a significant main effect of goal in either light-off or light-on trials (*p*-value < 0.025, corrected for multiple comparisons) were considered "goal-selective". For every goal-selective neuron, we asked in which goal arm the firing rate was overall higher (by comparing averaged firing rates across the three goal arm position bins described above) and defined this as the neuron's preferred goal. We then computed a goal index for each position in light-off and light-on trials, indicative of the difference in firing at the preferred versus nonpreferred goal. To this end, we used the linearized firing rates during outbound and inbound trajectories and while the animal was stationary at the goal, as described above, and calculated the goal index as follows:

$$Goal\ index\ =\ \frac{firing_{preferred\ goal}\ -\ firing_{nonpreferred\ goal}}{firing_{preferred\ goal}\ +\ firing_{nonpreferred\ goal}}$$

(4)

This measure was used to compare the strength of goal coding in light-off and light-on trials across positions.

## Goal decoding across linearized positions

Decoding of the visited goal arm in sample or choice phase was performed using a support vector machine classifier (Matlab function fitcsvm) trained on the activity of simultaneously recorded neuronal populations in each session. To this end, linearized firing rates were calculated separately for leftward and rightward trajectories, as described in the previous section, but for each trial separately. For each linear position bin and trial, we created goal activity vectors representing the firing rate of each recorded neuron in that position bin and trial. We only included sessions with more than 15 neurons with an average firing rate of >0.5 Hz in each phase. To estimate decoding performance under light-off conditions in the sample phase, we created 10 random pairs of sample left and sample right goal activity vectors from light-off trials and trained the classifier on nine left–right vector pairs (18 trials in total). The goals of the remaining pair of trials were then predicted based on the classifier. This was cross-validated 10 times, with each activity vector pair being used once for testing, and repeated for each linear position bin. The whole procedure was repeated 100 times, each time generating different pairs of trials. The results were averaged to obtain a measure of percent correct decoded goals for each session. For goal decoding in the choice phase, we were presented with the challenge that, depending on the animal's performance, some sessions contained more left choice trials than right choice trials, or vice versa. We therefore created random pairs of only eight left and eight right goal activity vectors and performed the decoding analysis as described above, which resulted in a slight reduction in overall decoding performance. Sessions with less than eight left or right choice trials were excluded from the analysis. To estimate decoding performance during HPC inactivation, we trained and tested the classifier as described before but now using light-on trials. Then, we compared the decoding accuracy in each phase and at each position in light-off versus light-on trials.

## Analysis of local field potentials and phase-locking of PFC spikes

To extract local field potentials (LFPs), broadband recordings in the PFC were first down-sampled to 2 kHz and low-pass filtered (Matlab function decimate, Chebyshev Type 1 filter). Spectral frequency estimates were then obtained by

convolving the LFP signal with a series of Morlet wavelets with center frequencies of 1–100 Hz and a length of three cycles, and then taking the absolute numbers of the wavelet transform. The resulting values represent the power fluctuation in the respective frequencies over time. For spectral analysis during optogenetic manipulations in the SWM task, the power of each frequency was averaged over the time the animal spent in the stem or goal arm during sample or choice outbound trajectories in light-off and light-on trials. To compare changes in the theta frequency band, power values were additionally averaged between 4 and 12 Hz.

Phase-locking of PFC neurons to the phase of local theta oscillations was analyzed by first filtering the LFP between 4 and 12 Hz using a zero-phase-delay filter with a linear phase function (Matlab function fir1) and extracting the phase of each filtered sample using the Hilbert transform. Every spike of PFC neurons was then assigned to the phase of theta oscillations happening closest in time. The strength of phase-locking for each neuron was assessed for the time the animal spent in the stem or goal arm during sample or choice outbound trajectories of light-off and light-on trials. We calculated the mean resultant vector length (MRL) as the sum of the unit vectors representing the phases at which each spike occurred, divided by the number of spikes. To ensure a representative estimate of spike phase, only neurons with at least 30 spikes per analyzed condition (stem sample light-off, goal arm sample light-off, stem sample light-on etc.) were included. To test whether a neuron was significantly phase-locked in any of the conditions, we used Rayleigh's test for uniformity of circular variance and a significance threshold of $p < 0.05$.

## Supporting information

**S1 Fig. Placement of optic fibers and recording electrodes. (A–B)** Placement of optic fibers in dHPC **(A)** and vHPC **(B)**. **(C)** Positions of recording electrodes in the PFC. Numbers indicate anteroposterior position relative to bregma. Colors indicate the placements of each animal and the symbols indicate the experimental group to which they belong. PrL, prelimbic cortex; IL, infralimbic cortex; ACC, anterior cingulate cortex. Atlas pictures are adapted from Franklin and Paxinos (2012) [87].
(TIF)

**S2 Fig. Analysis of behavioral performance and running speed. (A)** The number of training sessions required to reach criterion performance (left) and performance on the last training session (right) did not differ between the three experimental groups. **(B–D)** Effects of hippocampal silencing during the sample phase **(B)**, delay phase **(C)** and choice phase **(D)** on animals' running speed in the sample and choice phase. Dark colored bars show running speed during task phases in which light was delivered ('light-on phase'), lighter colored bars show running speed in task phases where light was not delivered ('light-off phase') and gray bars show running speed in trials without light delivery in any phase ('light-off trials'). Statistical analyses were only performed for task phases in which light was delivered (see section "Results" for more details). **(E)** Relationship between running speed and performance during light on and light off trials. Each dot represents the performance and median running speed in a single session, measured in the same task phase in which light was delivered. Lines represent the linear fit of performance onto running speed for light-off and light-on trials. Error bars indicate mean ± s.e.m. across animals. *$p < 0.05$, **$p < 0.01$, sign-rank test. The data underlying this figure can be found at https://doi.org/10.12751/g-node.ls2xxj.
(TIF)

**S3 Fig. Analysis of behavioral performance over testing days.** Relationship between performance during light-on and light-off trials and day of testing, shown separately for sessions in which light was delivered in the sample **(A)**, delay **(B)** and choice **(C)** phase. Each circle represents the performance of one animal during light-off and light-on trials during a single testing day. Lines represent the linear fits of performance onto testing day, separately for light-off and light-on trials. A light × day ANCOVA revealed a main effect of light for ArchT-dHPC sample light ($p < 0.001$, **A** left), ArchT-vHPC sample

light ($p < .05$, **A** middle) and ArchT-dHPC choice light ($p < 0.001$, **C** left) but no light × day interaction for these experimental conditions ($p = 0.92$, 0.57 and 0.62, respectively), suggesting that the effects of HPC silencing were stable over the testing days. A main effect of testing day was only observed for sample light delivery in the ArchT-vHPC group ($p < 0.05$, **A** middle) and choice light delivery in the GFP group ($p < 0.01$, **C** right). Note that the position of the individual data points was jittered slightly along the $x$-axis in order to improve their visibility (all analysis was performed on unjittered data). The data underlying this figure can be found at https://doi.org/10.12751/g-node.ls2xxj.
(TIF)

**S4 Fig. Identification and classification of single-units. (A)** Clusters of three single units in a 3-dimensional space defined by the first 3 principal components of waveform features. Unsorted spikes are shown in gray. **(B)** Spike autocorrelograms of the three single-unit clusters shown in **(A)**. **(C)** Waveforms of the clusters shown in **A** on the two channels (left and right) of the stereotrode. **(D)** Valley-to-peak and half-spike width of all recorded neurons. A 2-dimensional Gaussian mixture model was used to classify neurons as putative pyramidal (pPYR) and interneurons (pINT; see section "Methods"). Gray points indicate neurons with low classification confidence (see section "Methods"). **(E)** Normalized waveforms of pPYRs and pINTs. Shaded area represents the mean ± s.e.m across neurons. **(F)** Percentage of pPYR and pINTs in each experimental group. The data underlying this figure can be found at https://doi.org/10.12751/g-node.ls2xxj.
(TIF)

**S5 Fig. Effect of dHPC and vHPC silencing on firing rates of putative pyramidal neurons and interneurons. (A)** Percentages of excited, inhibited and non-modulated pPYRs (left) and pINTs (right) across experimental groups. The number of recorded neurons in each group is indicated at the bottom of the bars. **(B)** The percent change in firing rate during light delivery relative to baseline in excited (left) and inhibited (right) pPYRs ($n = 9$ excited and 11 inhibited in ArchT-dHPC, 58 excited and 44 inhibited in ArchT-vHPC group). Box plots represent the median (line), 25th and 75th percentiles (box) and the whiskers extend to the minimum and maximum values within 1.5 times the interquartile range below and above the 25th and 75th percentiles, respectively. **(C)**. Normalized firing rates (baseline subtracted) around light onset of excited (left) and inhibited (right) pPYR neurons in ArchT-dHPC and ArchT-vHPC mice. Shaded areas indicate mean ± s.e.m. across neurons ($n$ values as in **B**). **(D)** Response latencies of modulated pPYRs are longer in ArchT-dHPC mice ($n = 31$) than in ArchT-vHPC mice ($n = 86$). Error bars indicate mean ± s.e.m. across neurons. $^*p < 0.05$, Wilcoxon rank-sum test. The data underlying this figure can be found at https://doi.org/10.12751/g-node.ls2xxj.
(TIF)

**S6 Fig. dHPC and vHPC silencing effects on spatial firing patterns of prefrontal pyramidal neurons and interneurons. (A)** The percentage of PFC neurons significantly modulated by position in the sample phase (top) and the choice phase (bottom) is similar in light-off and light-on trials for pPYRs (left) and pINTs (right). **(B)** Spatial information (bits/spike) of PFC pPYRs (left) and pINTs (right) does not differ between light-off and light-on trials in either the sample phase (top) or the choice phase (bottom). **(C)** Correlation of linearized firing rates of prefrontal pPYRs (right) and pINTs (left) between light-off and light-on trials, separately for the task phase in which light was (dark colors) or was not (light colors) delivered. This revealed significant changes in spatial firing patterns of pPYRs when the dHPC is inhibited in the sample (top) or choice phase (bottom), or when the vHPC is inhibited in the sample phase (top). Box plots in **B** and **C** represent the median (line), 25th and 75th percentiles (box) and the whiskers extend to the minimum and maximum values within 1.5 times the interquartile range below and above the 25th and 75th percentiles, respectively. $^*p < 0.05$ and $^{**}p < 0.01$, Wilcoxon sign-rank test. Percentages in **A** and box plots in **B–C** were calculated over $n = 319$, 277, 185 pPYRs and 17, 75, 25 pINTs from ArchT-dHPC, ArchT-vHPC and GFP mice for sample light, and $n = 312$, 240, 188 pPYRs and 19, 67, 27 pINTs for choice light. The data underlying this figure can be found at https://doi.org/10.12751/g-node.ls2xxj.
(TIF)

**S7 Fig. Effect of hippocampal silencing on linearized firing rates. (A)** Correlations between linearized firing rates in even and odd light-off trials were similar for sample ('Sample Off') and choice phases ('Choice Off'), regardless of whether light was delivered in the sample (top) or the choice (bottom) phase. **(B)** For each task phase, correlations between even and odd light-off trials and between even and odd light-on trials ('Within light) were compared with correlations between even light-off and odd light-on trials and between even light-off and odd light-on trials ('Across light') for sample (top) and choice phase (bottom) silencing. These results confirm that linearized firing rates are altered by silencing of the dHPC and vHPC in the sample phase and by the dHPC in the choice phase (compare with Fig 4D). Box plots represent the median (line), 25th and 75th percentiles (box) and the whiskers extend to the minimum and maximum values within 1.5 times the interquartile range below and above the 25th and 75th percentiles, respectively. **$p < 0.01$, ***$p < 0.001$ sign-rank test. The data underlying this figure can be found at https://doi.org/10.12751/g-node.ls2xxj. (TIF)

**S8 Fig. Effect of HPC silencing on phase-locking of PFC neurons to local theta. (AB)**. Percentage of neurons significantly phase-locked to PFC theta oscillations (4–12 Hz) in the sample **(A)** and choice phase **(B)** during light-off and light-on trials. Results are shown separately for spikes recorded in the stem (left) and goal arms (right) of the T-maze in outbound trajectories. dHPC silencing in the sample phase decreased the percentage of phase-locked neurons in the goal arms of the T-maze (Fisher's exact test, $p < 0.05$). **(C,D)**. Strength of phase-locking to PFC theta oscillations, quantified as the mean resultant length (MRL) of phase angles, in the sample **(C)** and choice phase **(D)** during light-off and light-on trials. Results are shown separately for spikes recorded in the stem (left) and goal arms (right) of the T-maze. Box plots represent the median (line), 25th and 75th percentiles (box) and the whiskers extend to the minimum and maximum values within 1.5 times the interquartile range below and above the 25th and 75th percentiles, respectively, across neurons. **$p < 0.01$, Wilcoxon signed-rank test. **(E)** Power spectrum of LFP oscillations (left) in the PFC while animals were in the goal arm during sample phase outbound trajectories, either when the dHPC was inhibited (light on, green) or in light-off trials (grey). The theta frequency range (4–12 Hz) is indicated with a shaded rectangle. Lines show mean ± s.e.m. across sessions. Averaged PFC theta power in light-off and light-on conditions is shown on the right. Box plots represent the median (line), 25th and 75th percentiles (box) and the whiskers extend to the minimum and maximum values within 1.5 times the interquartile range below and above the 25th and 75th percentiles, respectively, across sessions. The data underlying this figure can be found at https://doi.org/10.12751/g-node.ls2xxj. (TIF)

## Acknowledgments

We thank Pascal Vogel, Beatrice Fischer, Jasmine Sonntag, Sebastian Betz and Natasha Khan for technical and histological assistance, Sevil Duvarci for comments on the manuscript and Johannes Hahn for code for data visualization.

## Author contributions

**Conceptualization:** Susanne S. Babl, Torfi Sigurdsson.

**Data curation:** Susanne S. Babl.

**Formal analysis:** Susanne S. Babl.

**Funding acquisition:** Torfi Sigurdsson.

**Investigation:** Susanne S. Babl.

**Methodology:** Susanne S. Babl.

**Project administration:** Torfi Sigurdsson.

**Software:** Susanne S. Babl.

**Supervision:** Torfi Sigurdsson.

**Visualization:** Susanne S. Babl.

**Writing – original draft:** Susanne S. Babl, Torfi Sigurdsson.

**Writing – review & editing:** Susanne S. Babl, Torfi Sigurdsson.

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
