## [Editor Report · Decision Letter 0]

28 Aug 2024

Dear Torfi,

Thank you for submitting your manuscript entitled "Distinct contributions of the dorsal and ventral hippocampus to spatial working memory and spatial coding in the prefrontal cortex" for consideration as a Research Article by PLOS Biology. First of all, please allow me to apologize for the long delay in getting back to you. I was hoping to receive advice from one of our academic editors, but this wasn't possible unfortunately due to the holiday period. 

So, your manuscript has now been evaluated by the PLOS Biology editorial staff and I am writing to let you know that we would like to send your submission out for external peer review. Please note that we were unable so far to find an academic editor to handle your manuscript, so we have not yet made a firm decision on whether the advance is sufficient for PLOS Biology. We will discuss this aspect with an academic editor once we have received the reviewer reports. 

Once your full submission is complete, your paper will undergo a series of checks in preparation for peer review. After your manuscript has passed the checks it will be sent out for review. To provide the metadata for your submission, please Login to Editorial Manager (https://www.editorialmanager.com/pbiology) within two working days, i.e. by Aug 30 2024 11:59PM.

Kind regards,

Christian

Christian Schnell, PhD

Senior Editor

PLOS Biology

cschnell@plos.org

---

## [Decision Letter · Decision Letter 1]

14 Oct 2024

Dear Dr Sigurdsson,

Thank you for your patience while your manuscript "Distinct contributions of the dorsal and ventral hippocampus to spatial working memory and spatial coding in the prefrontal cortex" was peer-reviewed at PLOS Biology. It has now been evaluated by the PLOS Biology editors, an Academic Editor with relevant expertise, and by several independent reviewers. 

In light of the reviews, which you will find at the end of this email, we would like to invite you to revise the work to thoroughly address the reviewers' reports.

As you will see below, the reviewers are overall supportive of publishing your manuscript and say that it is well done. Nevertheless, they raise a few concerns that we think need to be addressed, including clarifications, more methodological details and a few additional analyses.

Given the extent of revision needed, we cannot make a decision about publication until we have seen the revised manuscript and your response to the reviewers' comments. Your revised manuscript is likely to be sent for further evaluation by all or a subset of the reviewers.

**IMPORTANT - SUBMITTING YOUR REVISION**

*Re-submission Checklist*

*Published Peer Review*

*PLOS Data Policy*

*Blot and Gel Data Policy*

Sincerely,

Christian

Christian Schnell, PhD

Senior Editor

PLOS Biology

cschnell@plos.org

REVIEWS:

Reviewer #1: This is a clearly motivated, well designed, carefully analysed and eloquently described set of experiments. The authors are correct that previous studies of rodent HPC-PFC interactions have tended to focus on either dorsal or ventral HPC, so integrated study of the two HPC poles and their bearing on PFC physiology and function is useful. The study draws on longstanding use of the T-maze and well-established optogenetic silencing methods. While some effect sizes are small - particularly the effects of vHPC silencing on T-maze % correct (Fig 1D) - the analyses are logical and comprehensive. It is a shame that single mice where not used for both dHPC and vHPC silencing, as opposed to separate groups. Also a shame that we do not know the effects of silencing d/vHPC on neural activity in v/dHPC (since the results imply that silencing one may not effect the other, surprisingly). Nevertheless, the headline finding - of dissociable requirements for dorsal and ventral HPC activity during sample vs choice (but not delay) phases of the T-maze - offers useful food for thought and advances the field.

I have only minor comments:

1. To complement the spatial/goal decoding analyses, did silencing of either dHPC or vHPC impact phase-locking of PFC units to local theta? This would add useful detail to potential mechanisms through which silencing modulates PFC activity

2. In Fig 2C, it looks like units with higher baseline firing rates (which many would suspect as putative interneurons) are more likely to reduce firing during silencing. Units should be classified based on spike width and firing rate and putative pyramidal/interneuronal units analysed accordingly.

3. Was there any relationship between PFC unit response to HPC silencing and the layer from which that unit was recorded?

4. Line 273 subtitle just needs rephrasing I think - in general terms, vHPC alone is not sufficient for goal location encoding

5. Figs 5 B and C could swap positions, to keep presentation order consistent with panels D-I

6. Title: Are d/v contributions "distinct" or (more cautiously) "dissociable"?

Reviewer #2 (Antonio Fernandez-Ruiz): This study employed optogenetic manipulations and electrophysiological recordings in behaving mice to investigate the contributions of the dorsal and ventral hippocampus to spatial working memory performance and neural correlates in the prefrontal cortex (PFC). Despite much research on this topic, only few studies have attempted to dissociate the respective roles of the dorsal and ventral hippocampus, which greatly differ in the functional properties and connectivity with the PFC. Therefore, the central question of this study is timely and of relevance for a broad audience. By optogenetically silencing either hippocampal subregions, the authors found distinct behavioral deficits in specific phases of a classical spatial memory task. The two manipulation also have different effects in the firing dynamics of PFC neurons, with a predominant effect of ventral hippocampal silencing in spatial stability and goal coding in PFC. The manuscript is clear and well written, and the results presented support the authors interpretation. Both experimental and analytical methods employed are sound. While I do not have major issues to raise, there are several aspects that need improvement or clarification before publication. 

* I imagine it is possible that behavioral performance increased with training days during the period included in the analysis. A statistical test should be conducted to assess whether there is a significant effect of day in performance and if this affects the comparisons between conditions. 

* Some information about PFC unit analysis is missing. Were single units classified into excitatory and inhibitory cell types? This should be done, or otherwise justified, since both cell types may greatly differ in their spatial coding correlates. It would be also useful to include some plots in Supplemental about unit recording, identification and classification. Please provide the numbers of simultaneous units recorded and those that went into populations analysis (e.g., Bayesian decoding)

* In general, figure legends are missing important information. The n's of units, sessions, etc. that went into each analysis or plot should be specified in each figure legend. 

* The performance impairment due to silencing the hippocampus during the choice phase of the task is interpreted as "required for using the remembered information to make a behavioral decision". This is a reasonable interpretation but an alternative one, common in the literature, is to interpret this as a deficit in memory recall. It would be worth to mention it. 

* The procedure used to classify neurons as spatially modulated (ANOVA of linearized firing rates using 10 cm bins) is extremely permissive is different from standard procedures in the field. The rationale for using it should be discussed (e.g., the low spatial modulation of PFC neurons in general).

* In the analysis described in pages 7-8 is not stated that left and right trials are merged, although I assume that was the case, please clarify this in the text. The rationale for merging left and right trials in the analysis of Fig. 3 and 4 could use further justification, as it is not the most common way of doing these analyses. Would the results change in any meaningful way if analysis would be done independently for each trial type? 

* The effect size in Figure 4D looks very small. Effects could be partially obscured for the large dispersion of the data. A better way of visualizing them could be to also plot a normalized ratio of change per cell. 

* The results in Fig. 5 are quite interesting but require further clarification. Reading the text is not clear to me whether the scoring of cells' goal selectivity was done merging together inbound and outbound trials or these were considered independently. Was "goal selectivity" present in both outbound and inbound trials? If that was the case, more than goal selectivity is arm selectivity. But if it was only present in outbound trials, as the examples suggest, it would be more indicative of goal selectivity. This needs to be statistically assessed. Also related, was there direction selectivity (i.e., the same arm location in outbound versus inbound trials) in PFC firing? If it exists in control trials, was it affected by the manipulations? 

* The classification of cells as significantly excited or inhibited by light is too permissive (just higher or lower than baseline). This should be properly assessed by using a stricter criterion (e.g., mean +/- 2SD or 95% of shuffles). 

Reviewer #3: Babi and Sigurdsson examined the relative contributions of dorsal and ventral hippocampus to both working memory performance and mPFC activity. They found that only dorsal inactivations during the sample phase decreased performance, but optogenetically silencing during either sample or test phases of the task. They found few changes in unit or ensemble activity in PFC, which is notable given the past literature, even showing that positional decoding accuracy from PFC ensembles was not affected. They did show that vHPC inactivation during the sample phase did alter PFC goal differential activity. This supports the idea that vHPC has more influence over PFC activity than dHPC, but overall, much less influence in general than previous literature would lead one to believe. On the whole, this is a great paper. It is well written, well analyzed, and the data are clear. I do think that a few presentation and analysis related points need to be addressed to ensure that this paper has the impact on the literature that is should. 

Major Points:

The two most notable points for me in this article are not highlighted as they should be. 

1) Silencing either d or vHPC does not affect PFC spatial decoding. This is a key finding that is contrary to the beliefs of most researchers. This should be emphasized much more strongly. It should appear in the abstract and be discussed within the framework of the literature that has concentrated on neural interactions during this task (Jones and Wilson, 2005; Hyman et al., 2010; Hallock et al., 2016; Spellman et al., 2018; Benchenane et al., 2014; Jadhav et al., 2016). 

2) The story here with the behavior and recording results is that, dHPC is needed to perform the task (consistent with past findings), but vHPC is needed to pass along the important HPC info to PFC (goal location). I think the whole paper could be written more directly to get to this compelling narrative for which the data clearly shows. 

The behavioral analysis presented in Fig. 1 could be analyzed better. I see three possible better approaches, either embracing the lack of usage of repeated measures approaches or use a repeated measures approach. 1) For each task phase put all 6 sets of datapoints into a 2 X 3 ANOVA (light on/off X area group). 2) Put all task phases, light conditions, and area groups into one big 3 factor ANOVA. 3) If you are gonna analyze everything separately a paired ttest or other repeated measures approach would be more appropriate. Also, the difference score results comparing v and d HPC inactivation should be shown in the figure. I found myself already mentally calculating that from the available graphs. It's an important point and should be properly presented. Additionally, I'm not really convinced that the vHPC silencing is actually influencing behavior, since the light on percentage is equal to GFP performance. 

Minor Points:

L71 - this statement is not the whole story, multiple publications have shown that frontal cortex can lead the HPC under certain conditions such as object in place (Place et al., 2016), and during remote memory recall (Wirt & Hyman, 2019, Makino et al., 2019, and McCormick et al., 2019)

---

## [Decision Letter · Decision Letter 2]

28 Feb 2025

Dear Torfi,

Thank you for your patience while we considered your revised manuscript "Dissociable contributions of the dorsal and ventral hippocampus to spatial working memory and spatial coding in the prefrontal cortex" for publication as a Research Article at PLOS Biology. This revised version of your manuscript has been evaluated by the PLOS Biology editors, the Academic Editor and the original reviewers.

Based on the reviews and on our Academic Editor's assessment of your revision, we are likely to accept this manuscript for publication, provided you satisfactorily address the remaining points raised by the reviewers. Please also make sure to address the following data and other policy-related requests:

* We would like to suggest a different title so it complies with our journal style: "The dorsal and ventral hippocampus contribute differentially to spatial working memory and spatial coding in the prefrontal cortex"

* Please add the links to the funding agencies in the Financial Disclosure statement in the manuscript details.

* DATA POLICY:

Regardless of the method selected, please ensure that you provide the individual numerical values that underlie the summary data displayed in the following figure panels as they are essential for readers to assess your analysis and to reproduce it: 1EFHIKL, 2EG, 3CDGH, 4DEF, 5DEF, S2ABCDE, S3ABC, S5BD, S6ABC, S7AB and S8ABCDE

* CODE POLICY

We expect to receive your revised manuscript within two weeks. 

*Published Peer Review History*

*Press*

Sincerely,

Christian

Christian Schnell, PhD

Senior Editor

cschnell@plos.org

PLOS Biology

Reviewer remarks:

Reviewer #1: Thank you for your careful and considered responses to all 3 reviewers - you've done an excellent job.

Reviewer #2 (Antonio Fernandez-Ruiz): The authors have addressed all the issues raised in my original review. The manuscript has been substantially improved with new analysis and clarifications. I recommend this work for publication without further revisions needed. 

Reviewer #3: The authors have done an excellent job of addressing all of my concerns. This is a fantastic paper and will make a great contribution to the literature. In my initial review I misremembered the date of an important citation that belongs in this paper. It is McCormick et al 2020, where they show mPFC activation leads the HPC during memory recall in humans. Find the article here: https://academic.oup.com/cercor/article/30/11/5972/5860957.

---

## [Editor Report · Decision Letter 3]

1 Apr 2025

Dear Torfi,

Thank you for the submission of your revised Research Article "The dorsal and ventral hippocampus contribute differentially to spatial working memory and spatial coding in the prefrontal cortex" for publication in PLOS Biology. On behalf of my colleagues and the Academic Editor, Jozsef Csicsvari, I am pleased to say that we can in principle accept your manuscript for publication, provided you address any remaining formatting and reporting issues. These will be detailed in an email you should receive within 2-3 business days from our colleagues in the journal operations team; no action is required from you until then. Please note that we will not be able to formally accept your manuscript and schedule it for publication until you have completed any requested changes.

PRESS

Sincerely, 

Christian

Christian Schnell, PhD

Senior Editor

PLOS Biology

cschnell@plos.org